# Curcumin enhances bedaquiline's efficacy against *Mycobacterium abscessus*: *in vitro* and *in vivo* evidence

Dan Luo,[1,2] Weile Xie,[1,2] Zhe Wang[1,2]

**ABSTRACT** In this study, we describe the combined effects of bedaquiline (BDQ) and the natural product curcumin (CUR) on *Mycobacterium abscessus*. In both *in vitro* and *in vivo* experiments, CUR enhanced BDQ's inhibitory effect. This combination reduced *M. abscessus* survival under nutrient-deprived, hypoxic, and acidic conditions, accelerated ATP depletion, mitigated BDQ-induced respiratory compensation, and effectively improved infection outcomes in both normal and immunosuppressed mice. Metabolomics analysis revealed that adding CUR to BDQ exacerbated BDQ-dependent downregulation of purine and pyrimidine metabolism and amino acid synthesis. Thus, BDQ-CUR combination therapy could potentially be applied to treat *M. abscessus* infections.

**IMPORTANCE** *Mycobacterium abscessus* is an emerging pathogen that causes pulmonary infections, particularly in immunocompromised patients. It exhibits natural resistance to many anti-tuberculosis drugs, posing significant challenges for both patients and physicians, thereby raising the need for innovative drug discovery. Here, we describe the combined effects of bedaquiline (BDQ) and curcumin (CUR) on *M. abscessus*. *In vitro* and *in vivo* studies have shown that CUR enhances the inhibitory effect of BDQ. Additionally, we investigated the synergistic effects at the metabolic level. Thus, these findings highlight the potential of BDQ-CUR combination therapy against *M. abscessus* infections.

**KEYWORDS** *Mycobacterium abscessus*, bedaquiline, curcumin, synergy, combination

Non-tuberculous mycobacteria (NTM) cause chronic opportunistic pulmonary infections in susceptible populations (1). *Mycobacterium abscessus* is a prevalent pathogen among NTM species (2, 3). Treatment options are limited because of natural resistance to many commonly used antimicrobials. Current therapies, requiring 18–24 months of at least three drugs (4), yield poor outcomes; only 50% of *M. abscessus* and 70% of *Mycobacterium avium complex* (MAC) patients achieve prolonged culture conversion (5). This underscores the urgent need to develop new drugs for treating NTM infections.

Bedaquiline (BDQ) is an antibiotic approved for treating multidrug-resistant tuberculosis (MDR-TB). It inhibits the proton pump of mycobacterial ATP synthase, causing ATP depletion, unstable pH homeostasis, and cell death (6, 7). BDQ exhibits moderate-to-high *in vitro* activity against NTM (8–10) and demonstrates bacteriostatic effects in monotherapy in mouse models of *Mycobacterium avium* and *M. abscessus* infections (11, 12). According to Philley et al. (13), BDQ offers potential therapeutic benefits for patients with severe MAC and *M. abscessus* lung disease, making it a promising treatment option for these infections. Although adding BDQ to failing regimens for MAC and *M. abscessus* infections improves symptoms, it neither prevents microbiological failure nor inhibits the emergence of BDQ resistance (13, 14). Therefore,

**Peer Reviewers** Shail Mehta, Washington University in St. Louis, Saint Louis, Missouri, USA; Gabriela Hädrich, Universität Wien, Wien, Austria

Address correspondence to Zhe Wang, wangz@sjtu.edu.cn.

Dan Luo and Weile Xie contributed equally to this article. The order of authors was determined alphabetically.

The authors declare no conflict of interest.

See the funding table on p. 14.

it is necessary to combine BDQ with other drugs to enhance its efficacy and reduce the dosage of antibacterial agents.

Drug repurposing is a promising strategy to accelerate drug discovery. Curcumin (CUR), the principal curcuminoid derived from the plant *Curcuma longa*, has been extensively studied for its biological and chemical properties. CUR exhibits a broad spectrum of pharmacological properties, including antimicrobial, anti-inflammatory, antioxidant, and antitumor activities. Pharmacological studies have suggested that CUR exerts significant protective effects against TB (15). Kotwal et al. (16) explored the effects of plant-based natural products on the pharmacokinetics of BDQ and found that CUR increased the plasma concentration of BDQ when used in combination. This increase may be attributed to CUR's ability to improve BDQ absorption and slow its metabolism by inhibiting P-GP-mediated efflux. However, data on the effectiveness of the BDQ-CUR combination against *M. abscessus* are limited, especially *in vivo*. However, the mechanisms underlying the antibacterial activity of BDQ-CUR remain largely unknown. In this study, we evaluated the BDQ-CUR drug combination against *M. abscessus* both *in vitro* and *in vivo* and investigated the underlying molecular mechanisms. These findings present a novel treatment approach for *M. abscessus*.

## RESULTS AND DISCUSSIONS

### Curcumin enhances the efficacy of bedaquiline against *M. abscessus in vitro*

Time-kill assays were performed using the *M. abscessus* ATCC 19977 strain. As shown in Fig. 1A, the combination of CUR and BDQ produced sustained bacteriostatic effects. CUR monotherapy did not exhibit bactericidal effects. Although BDQ monotherapy (4 µg/mL) initially inhibited bacterial growth, regrowth was observed within 14 days. These results demonstrate that CUR effectively inhibits BDQ's *in vitro* growth inhibition against *M. abscessus*. This supports previous research showing that CUR acts not only as a potential antibiotic resistance breaker but also as an effective adjuvant therapy for BDQ (17).

### *In vitro* evaluation of BDQ-CUR combination

Similar to *Mycobacterium tuberculosis*, *M. abscessus* evades destruction by macrophages and neutrophils after colonization, resulting in granuloma formation and survival under harsh conditions, such as acidic environments, nutrient deprivation, and hypoxia (18, 19). Previous studies have shown that the sensitivity of *M. tuberculosis* to new candidate drugs varies depending on the physiological state of the cells (active or inactive) (20, 21). Specifically, this includes the non-replicating physiological state of *M. abscessus* under oxygen- and nutrient-deficient (PBS or single-nutrient) conditions (22) and its cellular state under acidic conditions (23). Accurately replicating these *in vivo* conditions *in vitro* is crucial for studying the infection process and evaluating bactericidal activity. Studies have indicated that under nutrient deprivation, BDQ exhibits bactericidal effects against *M. abscessus* (24). We assessed the bactericidal effect of the BDQ-CUR combination on *M. abscessus* by simulating a non-replicative state using a starvation model. The results showed that the BDQ-CUR combination significantly reduced the survival rate of *M. abscessus* compared to BDQ alone, with a CFU/mL decrease of 0.8 $\log_{10}$ (Fig. 1B). Additionally, we used the Wayne model and a pH range from 6.0 to 4.5 (decreasing by 0.5 increments) to evaluate the survival of *M. abscessus* under hypoxic and acidic conditions. Under hypoxic conditions, the BDQ-CUR combination effectively reduced *M. abscessus* survival, with a CFU/mL decrease of 0.75 $\log_{10}$ (Fig. 1C). *M. abscessus* exhibited high acid tolerance; although BDQ alone had some inhibitory effects, the combination with CUR was more effective, reducing growth by 0.8–1.25 $\log_{10}$ (Fig. 1D through G). The ability of the BDQ-CUR combination to maintain bactericidal activity under extreme conditions is particularly important and may improve therapeutic strategies against prolonged *M. abscessus* infections.

*M. abscessus* can resist intracellular destruction and establish infections, prompting us to evaluate the antibacterial activity of the BDQ-CUR combination in RAW264.7

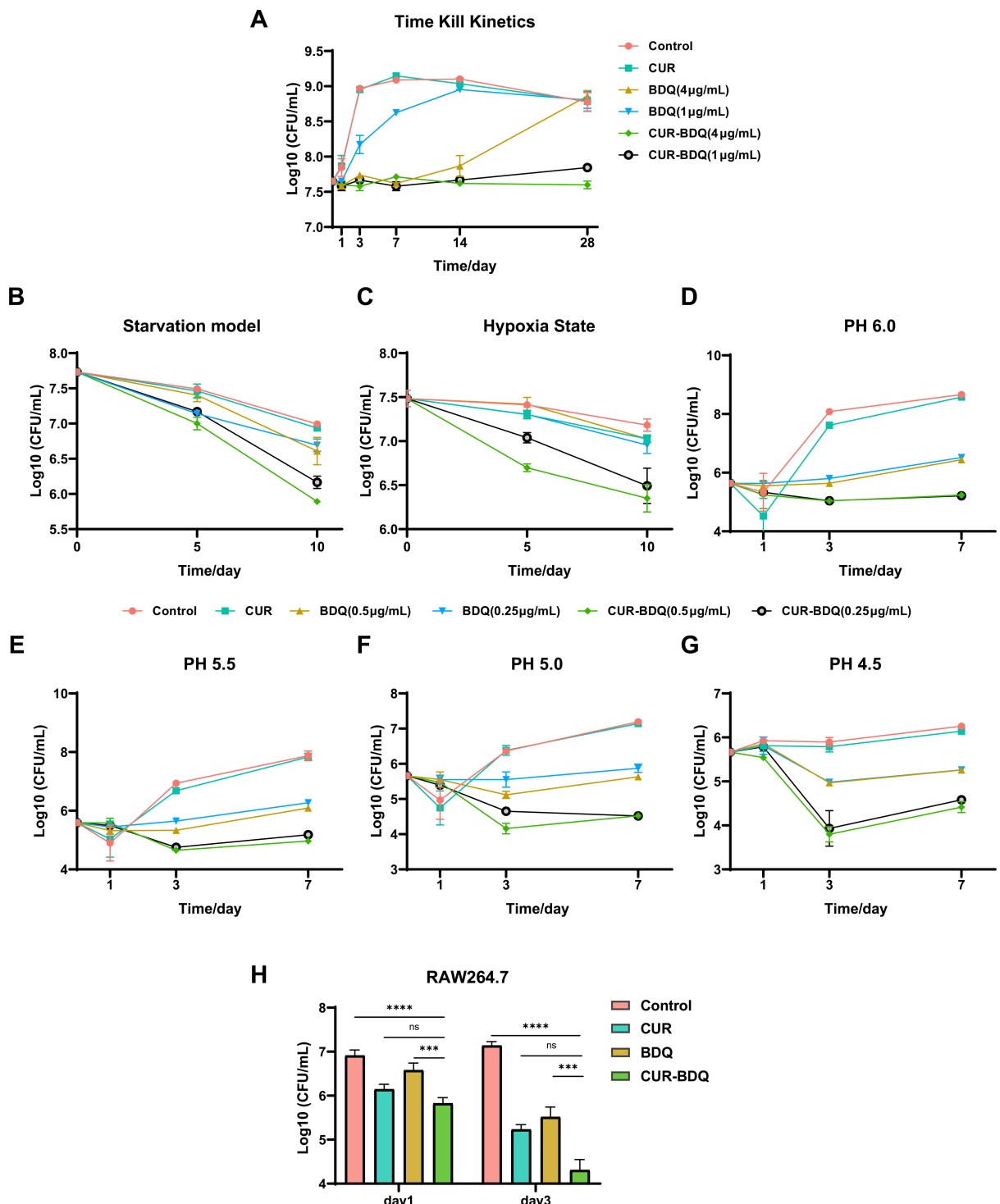

**FIG 1** Curcumin enhances bedaquiline's efficacy against *M. abscessus in vitro*. (A) Time kill kinetics: growth curves of *M. abscessus* treated with BDQ (1 or 4 µg/mL) and CUR (500 µg/mL), individually or in combination, compared to a DMSO-treated control, monitored over 28 days. Data represent mean ± SD from triplicate cultures. (B) Starvation model: survival of *M. abscessus* under nutrient-limited conditions treated with BDQ (0.5 or 0.25 µg/mL) and CUR (500 µg/mL), individually or in combination, monitored over 10 days. (C) Hypoxia state: survival of *M. abscessus* in hypoxic conditions treated with the same concentrations as in panel B, monitored over 10 days. (D–G) Acidic stress: survival of *M. abscessus* at pH levels 6.0, 5.5, 5.0, and 4.5 under the same treatments as in panel B, monitored over 10 days. (H) Macrophage infections: viability of *M. abscessus* in RAW264.7 macrophages treated with BDQ (1 µg/mL), CUR (500 µg/mL), or their combination (BDQ 1 µg/mL + CUR 500 µg/mL). CFU counts were determined at 1 and 3 days post-treatment.

macrophages. Within 3 days, the CFU/mL value of the BDQ-CUR combination in macrophages was reduced by at least 1 $\log_{10}$ compared with that of BDQ or CUR alone (Fig. 1H). Although CUR lacked *in vitro* inhibitory or bactericidal effects against *M. abscessus*, it exhibited superior antibacterial activity within macrophages compared to BDQ at an early stage. This was consistent with Bai et al.'s findings (25) in an *in vitro* human macrophage infection model, where CUR not only induced autophagy and apoptosis but also activated NF-κB, accelerating the clearance of *M. tuberculosis*. However, in the absence of macrophages, 50 μM CUR had no impact on *M. tuberculosis* growth (26–28). Therefore, it is crucial to identify drugs that exert antibacterial activity by modulating the host immune response. Such a strategy can reduce bacterial survival pressure and potentially delay the emergence of resistance to single drugs or combination therapies.

### *In vivo* evaluation of BDQ-CUR combination

Although the anti-*M. abscessus* effects of CUR and the CUR-BDQ combination in macrophages are promising, their protective effects in host organisms have not yet been confirmed in animal models. Kim and colleagues (29) demonstrated that C57BL/6 mice infected with *M. abscessus* ATCC 19977 could be a useful model for testing antimicrobials during infection development. As shown in Fig. 2A timeline and Fig. 2B results, BDQ-CUR could reduce the bacterial load in the lungs of normal C57BL/6 mice by day 7 of treatment. Compared to the untreated groups, BDQ-CUR (6/6), BDQ (3/6), and CUR (2/6) treatments reduced CFU counts by more than an order of magnitude (Table 1). HE staining revealed a reduction in inflammatory pathology in the lungs of BDQ-CUR-treated mice compared to untreated mice (Fig. 2C). Although the difference in lung bacterial load between the combined treatment group and BDQ alone was not statistically significant, the lung bacterial load showed a trend of 0.5 $\log_{10}$ unit reduction, and pathological sections showed significant improvement (Fig. 2B and C).

Furthermore, similar to the intranasal infection model, immunodeficient mice develop a persistent infection that reaches the spleen (systemic) (30, 31). We evaluated the effect of the combination using immunodeficient mice injected with cyclophosphamide (Fig. 2D) (32, 33). The results showed that after 3 days of treatment, the bacterial load in the BDQ-CUR groups was the same as the initial infection, while in other treatment groups (both the BDQ and CUR groups), the bacterial load significantly increased (Fig. 2E). The bacterial load in the spleen on day 3 in the combination group was also lower than that in the other groups, although it showed a slight increase relative to the initial infection (Fig. 2F). On day 7 of the combination treatment, compared with the BDQ group, both the lung and spleen bacterial loads significantly decreased by nearly 2 $\log_{10}$ CFU (Fig. 2E and F). Similarly, compared to the initial infection, the combination treatment resulted in a significant reduction in bacterial loads by nearly 2 $\log_{10}$ CFU in the lungs (4/4) and spleen (1/4) (Table 1). Pathological analysis on day 7 showed a reduction in the number of lymphocytes in these lesions, with moderate to minimal localized/focal histological features of inflammation observed in the BDQ-CUR groups. Severe and multifocal inflammation marked by the presence of lymphocytes and macrophages was observed in the lungs and spleen of the untreated group, in addition to extensive tissue damage (Fig. 2G). These findings support the reliability and effectiveness of the BDQ-CUR combination. Combination therapy has mitigated lung damage and drug toxicity from bacterial infections and may prevent the need to increase BDQ doses to combat drug-resistant *M. abscessus* (23, 34). Dose-escalating studies have indicated the safety of curcumin at doses as high as 12 g/day for over 3 months in humans (35). Previous results have shown that treatment with 16 or 32 μg/mL of CUR reduced the bacillary lung burden and improved survival rates in mice infected with drug-sensitive *M. tuberculosis* H37Rv (36, 37). Our results further confirm that CUR is a potential antimycobacterial drug against *M. tuberculosis* and *M. abscessus* infection. CUR may inhibit intracellular bacterial growth and promote the clearance of drug-sensitive strains by inducing caspase-3-dependent apoptosis and autophagy. This has been demonstrated in differentiated THP-1 human monocytes, primary human alveolar macrophages, and Raw 264.7 cells

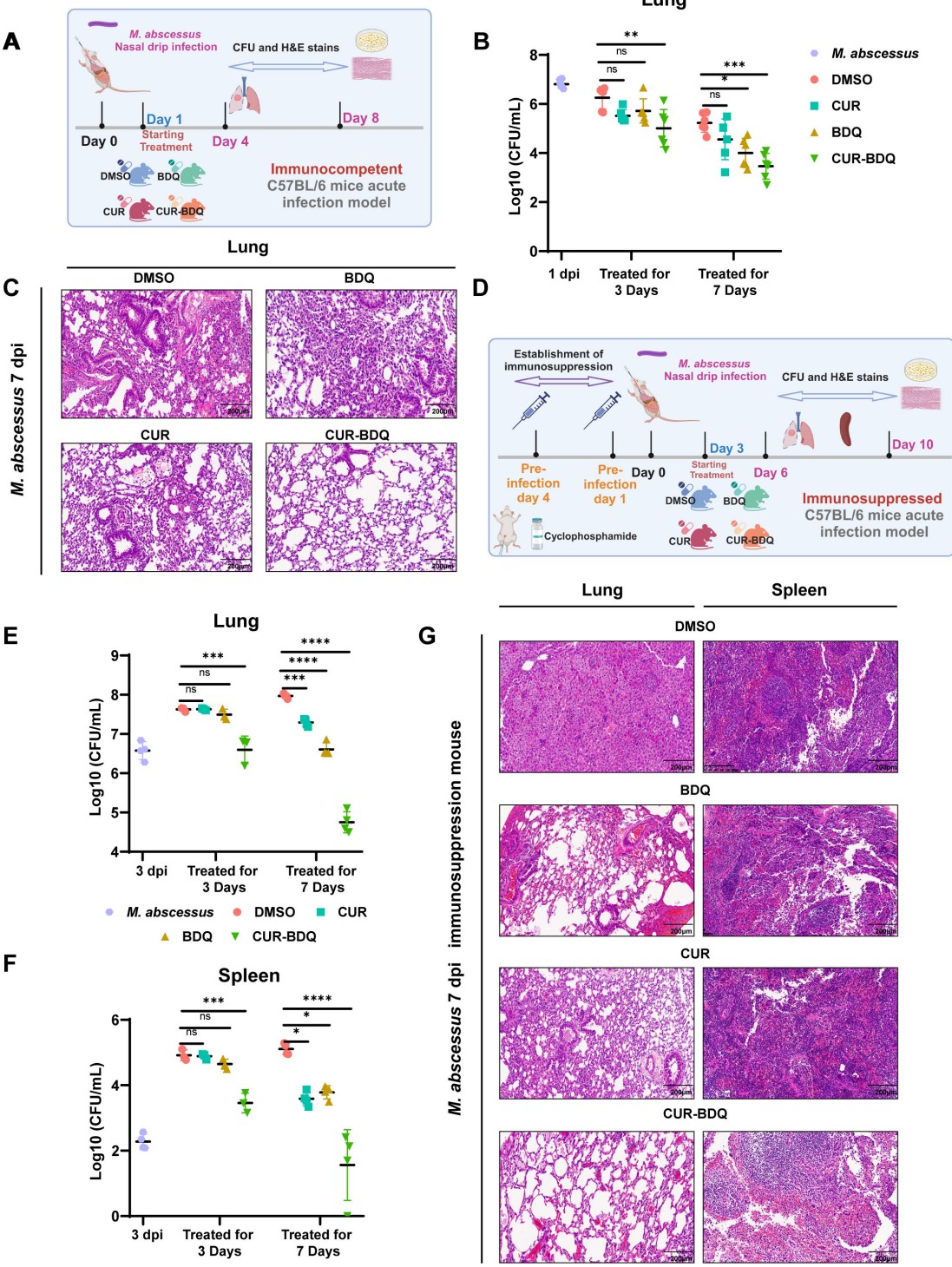

**FIG 2** Curcumin enhances bedaquiline's efficacy against *M. abscessus* in immunocompetent and immunosuppressed mouse models. (A) Schematic of the experimental procedure in immunocompetent mice: mice were intranasally infected with ~1 × 10⁹ CFU of *M. abscessus*. Six mice were sacrificed 1 day post-infection to determine the initial bacterial load, and the remaining mice were divided into four treatment groups: control (DMSO), CUR, BDQ, and CUR-BDQ. Drug treatment was administered via gavage, and mice were sacrificed at 3 and 7 days post-treatment (4 and 8 days post-infection) to evaluate lung bacterial loads. (B) Lung bacterial loads in immunocompetent mice treated once daily for 7 days with 30 mg BDQ/kg of body weight, 200 mg CUR/kg of body weight, or CUR-BDQ (30 mg BDQ/kg of body weight + 200 mg CUR/kg of body weight), or DMSO. (C). H&E staining of lung tissues from immunocompetent mice 7 days post-treatment. (D) Schematic of the experimental procedure in immunosuppressed mice: neutropenia was induced via intraperitoneal injection of cyclophosphamide (150 mg/kg of body weight) 4 days and 1 day prior to infection. Mice were then intranasally infected with ~1 × 10⁷ CFU of *M. abscessus*. Four

Fig 2 (Continued)

mice were sacrificed on day 3 post-infection to determine initial bacterial loads in the lung and spleen. The remaining mice were divided into the same four treatment groups as the immunocompetent model. Drug treatment was administered via gavage, and mice were sacrificed at 3 and 7 days post-treatment (6 and 10 days post-infection) to evaluate lung and spleen bacterial loads. (E and F) Lung and spleen bacterial loads in immunosuppressed mice treated once daily for 7 days with 30 mg BDQ/kg of body weight, 200 mg CUR/kg of body weight, CUR-BDQ (30 mg BDQ/kg of body weight +200 mg CUR/kg of body weight), or DMSO. (G) H&E staining of lung and spleen tissues from immunosuppressed mice 7 days post-treatment.

infected with *M. tuberculosis* H37Rv or MDR clinical isolates (25, 38). In addition, CUR may directly affect mycobacterial metabolic pathways, which are crucial for mycobacterial pathogenicity and host persistence (39). In summary, we demonstrated the efficacy of CUR in treating pulmonary infections in mice, highlighting its potential as an adjunctive agent to BDQ.

## Influence of BDQ-CUR on *M. abscessus* ATP flux and respiration

As BDQ exerts its antibacterial effects by targeting the ATP synthase c subunit and inhibiting ATP synthesis, we investigated the effect of BDQ alone and in combination with CUR on ATP flux in *M. abscessus*. Our results showed that BDQ significantly reduced ATP flux in *M. abscessus*, with the most pronounced depletion observed in the BDQ-CUR combination group, demonstrating a clear dose-dependent effect (Fig. 3A). Molecular docking studies have identified NAD-dependent DNA ligase as one of the predicted targets of CUR (15). To explore this interaction, we assessed the effect of CUR on $NAD^+$ levels and found that CUR significantly increased intracellular $NAD^+$ levels and the $NAD^+$/NADH ratio (Fig. 3B). NAD(H) homeostasis plays a crucial role in drug susceptibility and infection processes in mycobacteria (40–42). While $NAD^+$ depletion can induce lethal low-energy states, excessive $NAD^+$ accumulation elevates ROS levels, which can also be toxic. Oxidative stressors like $H_2O_2$ and HClO sharply increase $NAD^+$ levels and the $NAD^+$/NADH ratio, leading to lethal effects. Similarly, clofazimine treatment initiates an over-driven ROS cycle pathway, leading to an increase in $NAD^+$ and ROS, effectively killing mycobacterium (43–46). These challenges are often mitigated by mycobacteria through metabolic adaptations to withstand such stresses (43). *M. abscessus*, in particular, exhibits greater adaptability to ROS enrichment (47, 48). This adaptability may explain the lack of CUR-mediated inhibition observed in our study, as CUR-induced alterations in NAD(H) homeostasis appear insufficient to significantly impact its growth. Additionally, previous studies have reported a strong correlation between the bactericidal activity of ETC-targeted drug combinations and an increased intracellular $NADH/NAD^+$ ratio (49–51). Consistent with this, our results showed that the combination treatment led to a threefold increase in the $NADH/NAD^+$ ratio compared to the BDQ treatment alone. This increase was driven by NADH upregulation in the combination group (Fig. 3C and D). NAD(H) plays a critical role as a respiratory cofactor. Its imbalance initially triggers metabolic adaptation but ultimately leads to redox failure and cellular dysfunction (52–54). CUR enhanced BDQ disruption of redox balance and energy metabolism in *M. abscessus*, providing insights into the synergy of this drug combination.

Furthermore, to directly understand the respiratory effects of BDQ and CUR in *M. abscessuss*, both alone and in combination, we employed extracellular flux (XF) analysis to measure the oxygen consumption rate (OCR) of *M. abscessus* in real time as a marker of oxidative phosphorylation (OXPHOS) (Fig. 3E). We observed that the OCR of the BDQ-treated *M. abscessus* increased in a dose-dependent manner (Fig. 3F). Lamprecht et al. (55) reported that the BDQ-induced increase in respiration in *M. tuberculosis* was a specific response to ATP depletion in an attempt to restore energy homeostasis. As CUR affects the ETC electron flux, we anticipated a reduction in respiration. As expected, the OCR of CUR-treated cells decreased in a dose-dependent manner (Fig. 3G). Therefore, we investigated whether CUR could inhibit the feedback compensation mechanism induced by BDQ. Our findings indicated that the BDQ-CUR combination significantly reduced OCR

**TABLE 1** Summary of lung and spleen bacterial loads in immunocompetent and immunosuppressed mice after treatment (spleen samples collected only in immunosuppressed mice)

| Mouse model | Organ | Treatment groups | No. of improved infected mice (3 dpi)[a] | Mean difference from untreated group (log10)[b] | Mean difference from initial infection (log10) | No. of improved infected mice (7 dpi)[a] | Mean difference from untreated group (log10)[b] | Mean difference from initial infection (log10)[b] |
|---|---|---|---|---|---|---|---|---|
| Immunology normal | Lungs | CUR | 0/6 | 0.74 | – | 2/6 | 0.67 | – |
| | | BDQ | 1/6 | 0.54 | – | 3/6 | 1.23 | – |
| | | CUR-BDQ | 3/6 | 1.25[c] | – | 6/6 | **1.77** | – |
| Immunology suppressed | Lungs | CUR | 0/3 | –0.01 | –1.06 | 0/4 | 0.68 | –0.72 |
| | | BDQ | 0/3 | 0.13 | –0.91 | 0/4 | 1.36 | –0.03 |
| | | CUR-BDQ | 0/3 | **1.03** | –0.02 | 4/4 | **3.22** | **1.83** |
| | Spleen | CUR | 0/3 | 0.03 | –2.61 | 0/4 | 1.52 | –1.31 |
| | | BDQ | 0/3 | 0.27 | –2.37 | 0/4 | 1.32 | –1.51 |
| | | CUR-BDQ | 0/3 | **1.46** | –1.18 | 1/4 | **3.54** | **0.71** |

[a]Immunology normal mice model: for mice treated with BDQ, CUR, or the CUR-BDQ combination, *No. of infection improved mice* refers to the number of mice whose individual bacterial load decreased by at least 1 $\log_{10}$ unit compared to the average bacterial load of the *control group*, indicating an improvement in infection. Immunology suppressed mice model: in this model, *No. of infection improved mice* refers to the number of mice whose individual bacterial load decreased by at least 1 $\log_{10}$ unit compared to the average bacterial load of the *initial infection group*, indicating an improvement in infection.

[b]Mean difference from the untreated group or initial infection: calculated as the average bacterial load of the control or initial infection group minus the average bacterial load of the treatment group. A positive value indicates a reduction in the treatment group's bacterial load compared to the control or initial infection group; a negative value indicates an increase. "—" indicates that, in the immune normal mouse model, no comparison was made with the initial infection group.

[c]Bold values indicate the CUR-BDQ combination group, which is highlighted to emphasize its greater effectiveness in reducing bacterial load.

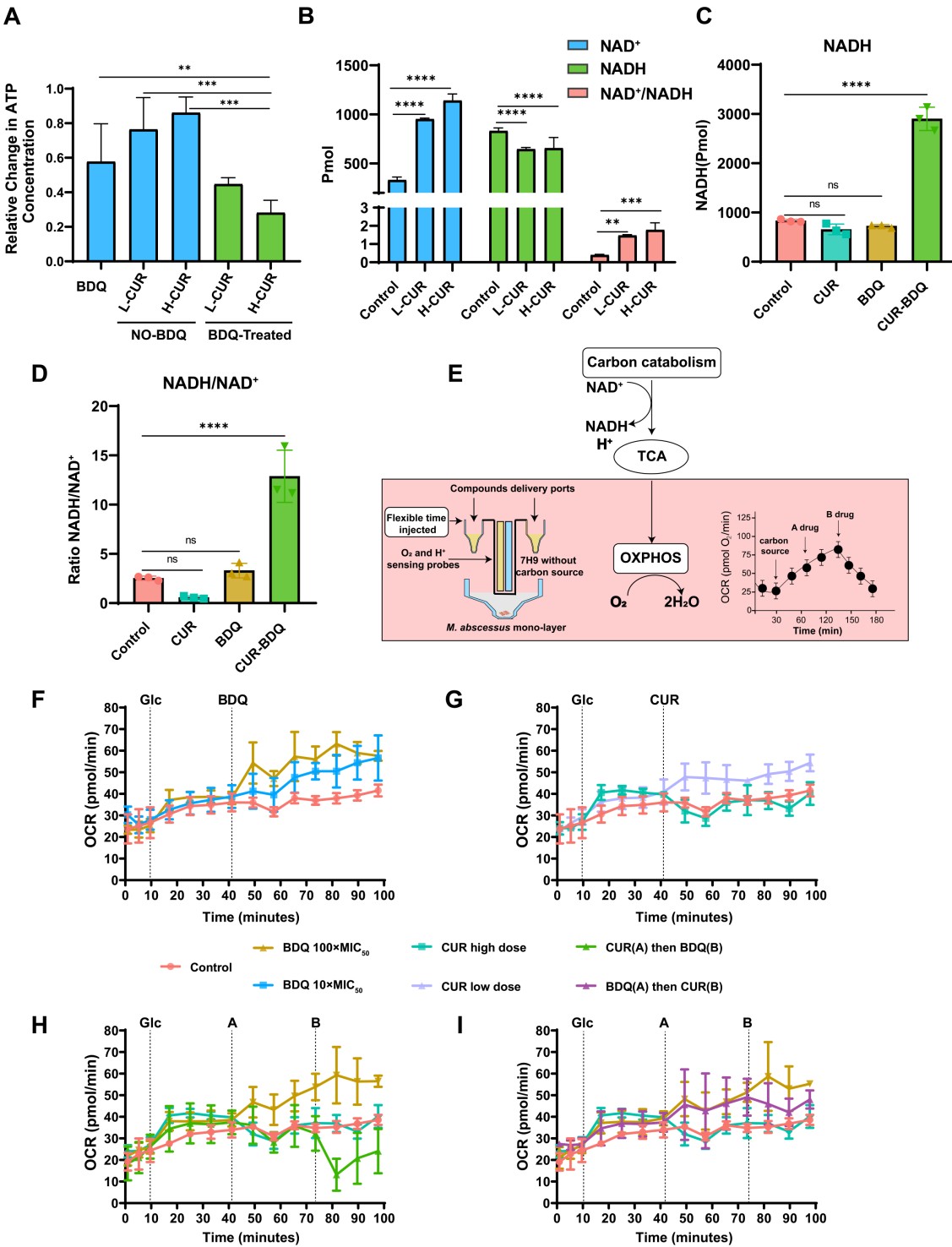

FIG 3 Impact of BDQ and CUR on ATP, NAD(H) metabolism, and cellular respiration in *M. abscessus*. (A) ATP content analysis in *M. abscessus* treated with BDQ (10 µg/mL, 10× MIC), CUR (200 and 500 µg/mL), or their combination for 4 hours. (B) Changes in NAD$^+$, NADH levels, and NAD$^+$/NADH ratio in *M. abscessus* treated with CUR at two concentrations (200 and 500 µg/mL) for 4 hours. (C and D) Changes in NADH levels and NADH/NAD$^+$ ratio in *M. abscessus* treated with BDQ (10 µg/mL), CUR (200 µg/mL), or CUR-BDQ combination (10 µg/mL BDQ + 200 µg/mL CUR) for 4 hours. (E) Schematic of the extracellular flux (XF) assay used to measure oxygen consumption rate (OCR) and related bioenergetic parameters. Drug compounds were introduced through ports, and dissolved O$_2$ and pH were continuously monitored. (F and G) OCR changes in *M. abscessus* upon exposure to BDQ (10× and 100× MIC) and CUR (200 and 500 µg/mL). (H and I) Sequential drug administration effects on OCR in *M. abscessus*: panel H shows CUR addition followed by BDQ, and panel I shows BDQ addition followed by CUR. Data were analyzed using Seahorse XF Wave software, with SD calculated from four biological replicates.

(Fig. 3H and I). Thus, the BDQ-CUR combination effectively inhibited *M. abscessus* growth by suppressing oxidative phosphorylation, thereby abolishing ATP production.

## Global metabolic response of *M. abscessus* to BDQ-CUR

To elucidate the downstream effects of ETC perturbation by BDQ, CUR, or their combination, we employed a metabolomics approach. Using the UPLC-Q-TOF/MS system, we analyzed perturbations in the metabolite spectrum of *M. abscessus*, considering the intricate interplay between multiple feedback loops and the flexibility of the ETC, which may result in complex and unexpected responses. In the PLSDA model, all replicates in each group were grouped into the same cluster. The CUR group and the untreated group (negative control, NC) showed clear aggregation behavior, whereas the BDQ-CUR (BC) group was more similar to the BDQ group (Fig. 4A). Among them, 37 metabolites were significantly downregulated after CUR treatment, 80 metabolites after BDQ treatment, and 97 metabolites after BC treatment (Fig. 4B). There were 37 differentially expressed metabolites (DEMs) that were common to all three groups, which mainly participated in purine and pyrimidine metabolism (Fig. 4C and D). This observation is consistent with previous studies in which purine and pyrimidine biosynthesis were the most recurrently affected metabolites among different drug treatments, playing an important role in the early drug-produced stress response. Moreover, KEGG enrichment analysis showed that the DEMs in the BC and BDQ groups were mainly involved in carbohydrate and energy metabolism, such as the citrate cycle (TCA cycle), pentose phosphate, propanoate pathway, and oxidative phosphorylation (Fig. 4D).

We next explored the trends in the DEMs between the BDQ-CUR combination and BDQ alone. These trends were consistent but more pronounced in the BC group. Specifically, we observed that dGDP, ADP, UDP-glucose, alanine, guanosine, and glutamine were downregulated by more than fourfold in the BC group compared to those in the BDQ group (Fig. 4E). This substantial downregulation of key metabolic intermediates disrupted nucleotide, carbohydrate, amino acid, and energy metabolism pathways (Fig. 4F and G). Our previous study had identified glutamine synthetase as a metabolic vulnerability factor in *M. tuberculosis* in response to BDQ, with a strong correlation between glutamine and ATP levels (56). Furthermore, we confirmed that *M. abscessus* treated with BDQ-CUR combination exhibited a low metabolic state, reduced respiration rate, and reduced macromolecular synthesis (Fig. 5). This observation aligns with other reports indicating that MTB remodels its metabolism to compensate for reduced ATP levels, thereby facilitating redox balance (57–60). Mackenzie et al. (49) found that BDQ reprogrammed *M. tuberculosis* metabolism, rendering it vulnerable to the genetic disruption of glycolysis and gluconeogenesis, leading to rapid sterilization when combined with OXPHOS inhibition. This mechanism may explain why the combination of BDQ and CUR effectively inhibits the growth of metabolically fragile strains of *M. abscessus*. Moreover, the inhibition of ATP synthesis by BDQ leads to the differential inhibition of various ATP-dependent metabolic processes. This "extension mechanism" aligns with emerging concepts that modifying the primary targets of antibiotics triggers a cascade of events within bacteria, ultimately disrupting cellular processes (61). Furthermore, we demonstrated that the combination of CUR and BDQ enhanced these effects.

## Conclusion

In summary, our research indicates that the combination of BDQ and CUR not only inhibits the growth of *M. abscessus* but also exhibits potential therapeutic effects. These effects include killing dormant bacteria, enhancing bacterial clearance from the body, reducing bacterial ATP synthesis and cell respiration, and causing significant metabolic reprogramming and cessation of macromolecular synthesis. Although these results are promising, further clinical studies are essential to validate the safety and efficacy of

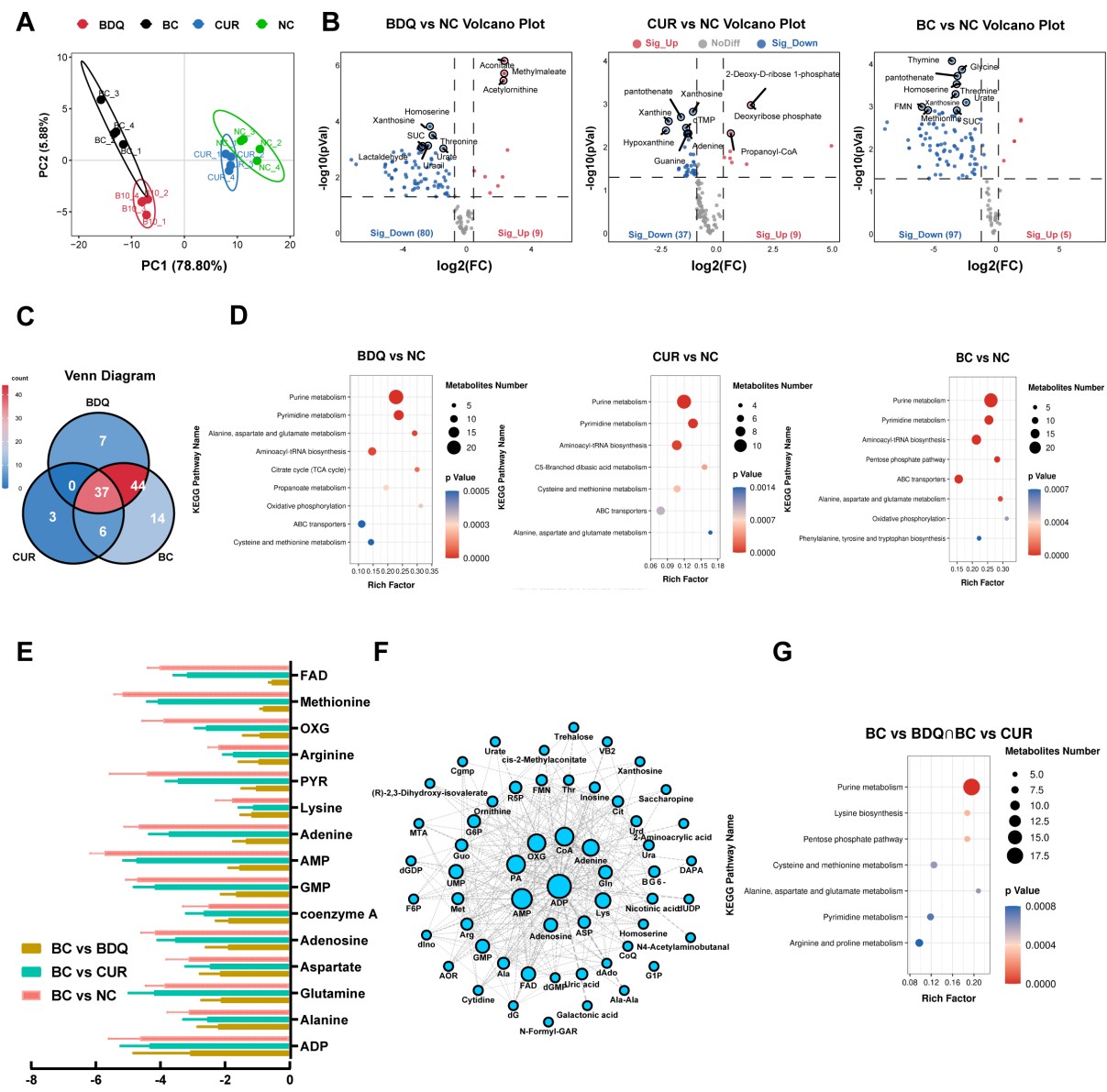

**FIG 4** Metabolic responses of *M. abscessus* to BDQ-CUR combination treatment. (A) PLS-DA analysis of metabolite profiles in *M. abscessus* treated with DMSO (NC), BDQ, CUR, and their combination (BC). (B) Volcano plots comparing metabolite expression in *M. abscessus* under BDQ, CUR, and BC treatments to the NC group. Significant differential metabolites (DEMs) are highlighted ($P \leq 0.05$, |fold change| $\geq 1.5$), with blue dots representing downregulated metabolites and red dots representing upregulated metabolites. (C) Venn diagram showing the overlap of DEMs identified in *M. abscessus* under BDQ, CUR, and BC treatments. (D) KEGG pathway enrichment analysis of DEMs under BDQ, CUR, and BC treatments compared to the NC group. Key pathways are visualized ($P \leq 0.001$). (E) Bar graph depicting $\log_2$ fold changes of key metabolites under BC treatment compared to the NC and single-drug treatments (BDQ or CUR). (F) Network diagram visualizing key DEMs, with blue nodes indicating downregulated metabolites. Node size reflects connectivity. (G) KEGG pathway enrichment analysis highlighting common pathways between BC and single-drug treatments (BDQ or CUR).

combination therapy in humans. This study provides a crucial theoretical foundation for the development of more effective treatment strategies for *M. abscessus* infections.

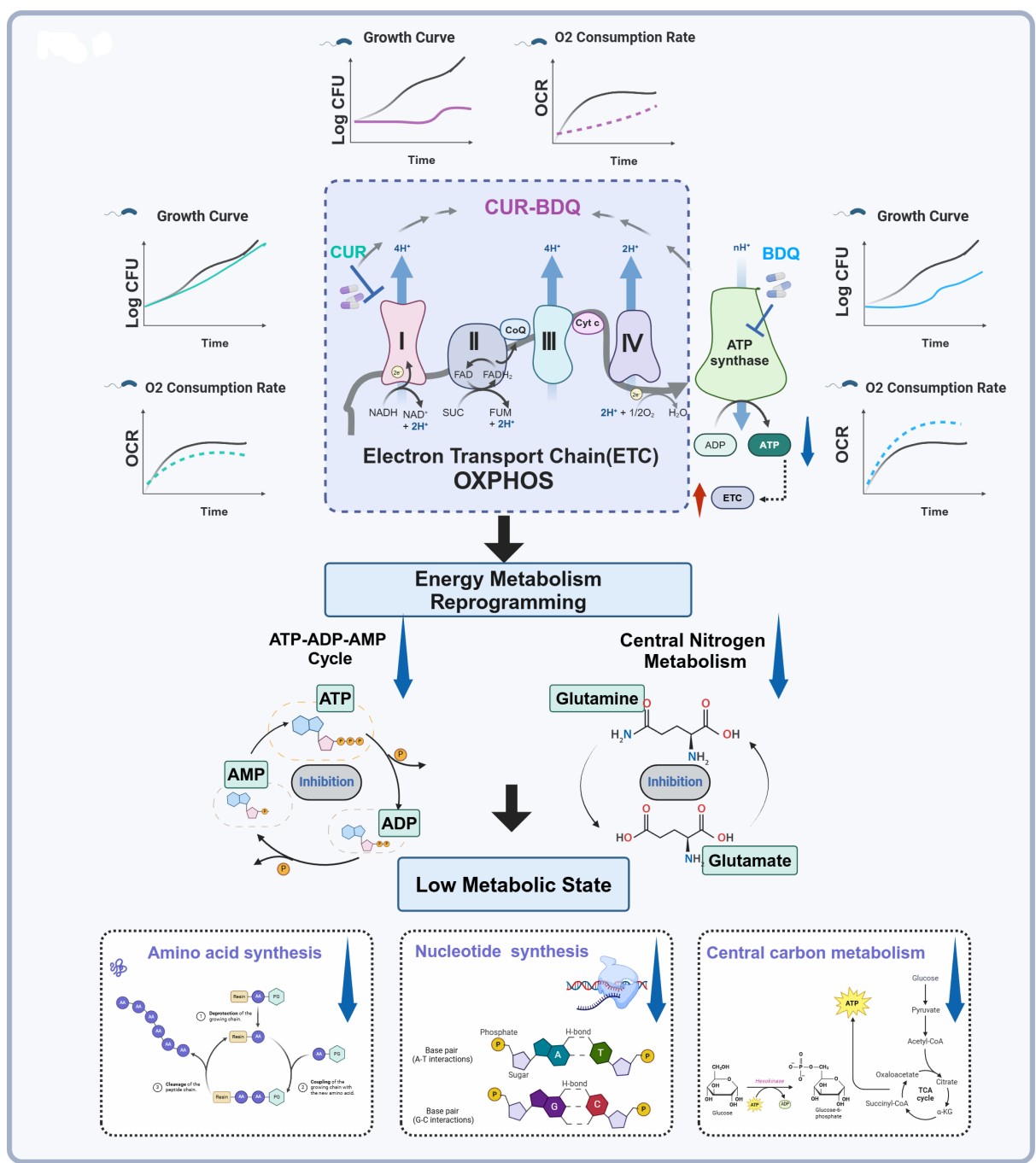

**FIG 5** Metabolic effects induced by BDQ-CUR combination in *M. abscessus.* This schematic illustrates the distinct and combined effects of BDQ and CUR on the metabolic state of *M. abscessus*. BDQ primarily targets ATP synthase, altering respiratory chain activity and enhancing cellular respiration as a compensatory mechanism. CUR treatment is associated with increased NAD$^+$ levels and an elevated NAD$^+$/NADH ratio, which contribute to impaired cellular respiration. The BDQ-CUR combination disrupts these compensatory mechanisms, leading to significant reductions in ATP production and metabolic activity, as well as an increase in the NADH/NAD$^+$ ratio. This synergistic effect enhances BDQ's mechanism of action by targeting key metabolic pathways critical to its efficacy, including the ATP-ADP-AMP cycle, nitrogen metabolism (glutamine and glutamate), amino acid synthesis, nucleotide synthesis, and central carbon metabolism, significantly extending the antimycobacterial efficacy of BDQ. This figure was created using BioRender.com.

## MATERIALS AND METHODS

### Bacterial strains, media, and reagents

*M. abscessus* (laboratory ATCC 19977 stains, resuscitated from a frozen stock) was grown in Middlebrook 7H9 broth (BD, USA) supplemented with 10% acid–albumin–dextrose–catalase or on 7H10 agar containing 0.5% glycerol and 10% oleic acid–albumin–dextrose–catalase (BD, USA). Cultures were grown at 37°C under aerobic conditions. The compounds bedaquiline, curcumin, and kanamycin (KM) were purchased from RHAWN Co., Ltd (Shanghai). BDQ and CUR were dissolved in a solution containing 20% (vol/vol) DMSO in water, while KM was dissolved in ultrapure water. All solutions were sterilized using a 0.22 μm filter before use.

### Time-kill curve assays

Time-kill curve assays were conducted using *M. abscessus* ATCC 19977. An overnight culture of ATCC 19977 was diluted in 10 mL of 7H9 broth to reach a final concentration of approximately $10^7$ CFU/mL. BDQ at 4 or 1 μg/mL and CUR at 500 μg/mL were added individually or in combination. The same volume of 20% DMSO was used as a control. Bacterial survival was monitored by collecting samples at the indicated time points (0, 1, 3, 7, 14, and 28 days) for CFU determination.

### Evaluation of drug combination effects in dormancy *M. abscessus*

*M. abscessus* stock cultures were grown in 7H9 medium to an $OD_{600}$ of around 1. The bacterial suspensions were washed three times in PBS with 0.05% Tween 80 as follows: cultures were centrifuged at 1,900 rpm for 10 minutes, the supernatant was removed, and cells were resuspended to the original volume in PBS with 0.05% Tween 80. After the third wash, the resuspended bacteria were incubated under nutrient starvation conditions at 37°C for the indicated duration. Bacterial survival was monitored by removing samples and culturing for CFU determination at 5 and 10 days.

For acidic pH stress, the pH of 7H9 medium was adjusted to 4.5, 5.0, 5.5, or 6.0 using HCl and then filter sterilized with a 2 μm filter. After exposure to acid stress conditions, samples were diluted and plated on 7H10 agar for bacterial enumeration.

For hypoxic dormancy, a modified Wayne's model, as described previously, was used (62). Methylene blue was added to 7H9 medium as an oxygen indicator, and flasks were sealed with rubber stoppers to prevent oxygen access. Cultures were incubated under hypoxic conditions for 5 or 10 days, after which the flasks were opened, and the cultures were serially diluted and plated on 7H10 agar to determine CFU/mL.

Under all experimental conditions, BDQ at 0.5 or 0.25 μg/mL and CUR at 500 μg/mL were added individually or in combination. The same volume of 20% DMSO was used as a control.

### Cellular ATP level and NAD(H) measurement in *M. abscessus*

Intracellular ATP and NAD(H) levels in *M. abscessus* ATCC 19977 were measured using the Enhanced ATP Assay Kit and the $NAD^+$/NADH Assay Kit (both from Beyotime, China). Bacteria were cultured in 7H9 broth to an $OD_{600}$ of approximately 1 and then diluted to a final concentration of $10^6$ CFU/mL. Cultures were treated with CUR (500 or 200 μg/mL), BDQ (10 μg/mL), their combination, and 20% DMSO and incubated at 37°C for 4 hours. After incubation, cultures were centrifuged at 12,000 rpm for 5 minutes at 4°C to collect bacterial pellets. For ATP measurement, the pellets were lysed with ATP lysis buffer and mechanically disrupted with 0.1 mm silica beads using a tissue homogenizer (Precellys, France) at 4°C. For NAD(H) measurement, the pellets were resuspended in $NAD^+$/NADH extraction buffer and disrupted under the same conditions. After disruption, the lysates were centrifuged, and the supernatants were collected. ATP levels were quantified by mixing the supernatant with ATP detection solution in a 96-well plate, incubating at room temperature for 2 minutes, and measuring luminescence using a BioTek microplate

reader (Agilent, USA). NAD(H) levels were determined by adding 20 μL of supernatant and 90 μL of working solution to a 96-well plate, incubating at 37°C for 20 minutes, and measuring absorbance at 450 nm using an enzyme-linked immunosorbent analyzer (AMR-100, Allsheng, China). All measurements were performed in three independent biological replicates.

## Measurement of oxygen consumption rate in *M. abscessus*

Overnight-cultured *M. abscessus* cells were centrifuged and washed twice with minimal medium lacking carbon sources. The cells were diluted to an $OD_{600}$ of 0.0015 in a carbon-free 7H9 medium. A total of 180 μL of the diluted cell suspension was added to XF cell culture microplates pre-coated with poly-D-lysine to promote cell adhesion. Baseline oxygen consumption rate was taken prior to antibiotic treatment using a Seahorse XFe96 Analyzer (Agilent, USA), following the manufacturer's instructions. Baseline OCR was measured over three cycles, each lasting 4 minutes. For the metabolic assay, glucose was automatically injected into each well at a concentration of 2 mg/mL using the A ports of the sensor cartridge. Antibiotic treatments, including BDQ (100 and 10 μg/mL), CUR (500 and 200 μg/mL), 20% DMSO (as a vehicle control), and carbon-free 7H9 medium (as a negative control), were administered sequentially through the remaining ports. The OCR was then measured for four cycles of 7 minutes each following each antibiotic injection.

## LC-MS-based metabolomics analysis of *M. abscessus*

Metabolomic profiling of *M. abscessus* was conducted following established methodologies (49). Bacteria were treated with BDQ (10 μg/mL), CUR (500 μg/mL), the combination of CUR-BDQ (1 μg/m L + 500 μg/mL), or the negative control (equal volume of 20% DMSO) for 1 day. Metabolites were separated using a Vanquish UHPLC system coupled with a Q Exactive Plus Mass Spectrometer (Thermo Fisher Scientific, USA). Metabolite identification was performed using TraceFinder software (Thermo Fisher Scientific, USA). DEMs were identified based on an absolute fold change (|fold change|) of ≥1.5 and *P*-value ≤ 0.05. Statistical analyses and data visualization were executed using R software (version 3.6.3). The raw liquid chromatography-mass spectrometry data have been deposited in the EMBL-EBI MetaboLights database under the identifier MTBLS10177.

## Evaluation of drug combination treatment effects of *M. abscessus* infection in RAW 264.7 macrophages

RAW 264.7 macrophages were seeded in 6-well culture plates at a density of $1 \times 10^6$ cells per well. Before infection, the cells were washed three times with DMEM to remove nonadherent cells. The macrophages were then infected with *M. abscessus* at a multiplicity of infection of 1 for 4 hours. After infection, the cells were washed twice with prewarmed 1× PBS and treated with DMEM medium containing 500 μg/mL KM for 1 hour to eliminate non-internalized bacteria. The cells were then washed with 1 mL PBS, and 2 mL DMEM medium containing BDQ (1 μg/mL), CUR (500 μg/mL), a combination of BDQ (1 μg/mL) and CUR (500 μg/mL), or the same volume of 20% DMSO was added to the respective wells. The cells were then incubated for 24 or 72 hours. At the specified time points, the cells were lysed using sterile PBS containing 0.025% SDS. The lysates were serially diluted and plated on 7H10 agar plates in triplicate. CFU was calculated to determine the bacterial load.

## *In vivo* treatment evaluation

*M. abscessus* infection was prepared by diluting the bacterial culture with PBS to a final volume of 20 μL per mouse. A total of 54 immunocompetent C57BL/6 mice were intranasally inoculated with *M. abscessus* (~$1 \times 10^9$ CFU/mouse). After 1 day, the infected mice were divided randomly into four treatment groups as follows: Group 1 received 20% DMSO only as a control. Group 2 was treated with BDQ at a dose of 30 mg/kg of

body weight. Group 3 received CUR at a dose of 200 mg/kg of body weight. Group 4 was treated with a combination of BDQ (30 mg/kg of body weight) and CUR (200 mg/kg of body weight). All treatments were prepared in 20% DMSO and administered by daily oral gavage for 1 week. Each mouse received 100 µL of the drug via intragastric administration.

On day 1 post-infection, six mice were sacrificed to determine the initial bacterial load in the lungs. On days 3 or 7 post-treatment, six mice from each group were sacrificed. The upper lobes of the right lungs were dissected, fixed in 4% paraformaldehyde for 24 hours, and sectioned for histological analysis using H&E staining. The remaining lung tissues were homogenized, serial diluted 10-fold, and plated on 7H10 agar. Bacterial load in the lungs was determined by counting colonies after 5 days of incubation at 37°C.

For the immunosuppression mouse infection, studies were conducted to mimic respiratory infection as described previously (33). Briefly, 32 6-week-old female C57BL/6 mice were rendered neutropenic by intraperitoneal injection of cyclophosphamide at 150 mg/kg of body weight, administered 4 and 1 days prior to infection. The mice were then infected intranasally with ~$1 \times 10^7$ CFU of *M. abscessus*. On day 3 post-infection, four mice were sacrificed to determine the initial bacterial load in the lungs and spleens. The remaining 28 infected mice were divided randomly into the same four treatment groups as the immunocompetent mice. Three or four mice from each group were sacrificed on days 3 or 7 post-treatment. Lungs and spleens were collected for CFU count and histological analysis using H&E staining.

## Statistical analysis

All data were presented as the mean ± SD from a minimum of three independent experiments. Statistical analyses were conducted using GraphPad Prism software version 8.0.1. Differences were considered statistically significant at *$P < 0.05$, **$P < 0.01$, ***$P < 0.001$, and ****$P < 0.0001$. The notation "ns" indicates a lack of statistical significance.

## ACKNOWLEDGMENTS

We thank Prof. Kyu Y. Rhee (Weill Cornell Medical College) and all members of the Wang lab for critical discussion and document revision.

This research was kindly supported by a grant from the National Key Research and Development Plans of China (No. 2021YFD1800401) to Z.W., National Natural Science Foundation of China (No. 32070128) to Z.W., and Shanghai Biomedical Science and Technology Support Special Project (No. 21S11900200) to Z.W.

## AUTHOR AFFILIATIONS

[1]Shanghai Key Laboratory of Veterinary Biotechnology, School of Agriculture and Biology, Shanghai Jiao Tong University, Shanghai, Shanghai, China
[2]Collaborative Innovation Center of Agri-Seeds, School of Agriculture and Biology, Shanghai Jiao Tong University, Shanghai, Shanghai, China

## AUTHOR ORCIDs

Dan Luo http://orcid.org/0000-0001-7678-951X
Zhe Wang http://orcid.org/0000-0002-7235-3046

## FUNDING

| Funder | Grant(s) | Author(s) |
| --- | --- | --- |
| MOST | National Key Research and Development Program of China (NKPs) | 2021YFD1800401 | Zhe Wang |
| MOST | National Natural Science Foundation of China (NSFC) | 32070128 | Zhe Wang |

| Funder | Grant(s) | Author(s) |
|---|---|---|
| SHANGHAI BIOMEDICAL SCIENCE AND TECHNOLOGY SUPPORT SPECIAL PROJECT | 21S11900200 | Zhe Wang |

## AUTHOR CONTRIBUTIONS

Dan Luo, Data curation, Formal analysis, Investigation, Methodology, Software, Validation, Visualization, Writing – original draft | Weile Xie, Data curation, Formal analysis, Investigation, Methodology, Validation, Visualization, Writing – original draft | Zhe Wang, Conceptualization, Data curation, Formal analysis, Funding acquisition, Investigation, Project administration, Resources, Supervision, Validation, Writing – original draft, Writing – review and editing

## ETHICS APPROVAL

All procedures were conducted in accordance with the Guidelines of the Animal Care and Use Committee of Shanghai Jiao Tong University. The animal study protocols were approved by the Institutional Animal Care and Use Committee of Shanghai Jiao Tong University.

## ADDITIONAL FILES

The following material is available online.

### Supplemental Material

**Supplemental material (Spectrum02295-24-s0001.docx).** Fig. S1 to S6; Table S1.

### Open Peer Review

**PEER REVIEW HISTORY (review-history.pdf).** An accounting of the reviewer comments and feedback.

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
