## [Reviewer comments · Microbiology Spectrum]

Microbiology Spectrum

Curcumin Enhances Bedaquiline's Efficacy Against *Mycobacterium abscessus*: In Vitro and In Vivo Evidence

Dan Luo, Weile Xie, and Zhe Wang

Corresponding Author(s): Zhe Wang, Shanghai Jiao Tong University

Review Timeline:

Submission Date:	September 12, 2024
Editorial Decision:	November 7, 2024
Revision Received:	January 2, 2025
Editorial Decision:	January 23, 2025
Revision Received:	January 27, 2025
Accepted:	February 2, 2025

Editor: Kayvan Zainabadi

Reviewer(s): Disclosure of reviewer identity is with reference to reviewer comments included in decision letter(s). The following individuals involved in review of your submission have agreed to reveal their identity: Shail Mehta (Reviewer #1); Gabriela Hädrich (Reviewer #2)

Transaction Report:

DOI: <https://doi.org/10.1128/spectrum.02295-24>

Re: Spectrum02295-24 (**Curcumin Enhances Bedaquiline's Efficacy Against *Mycobacterium abscessus*: In Vitro and In Vivo Evidence**)

Dear Prof. Zhe Wang:

Thank you for the privilege of reviewing your work. Below you will find my comments, instructions from the Spectrum editorial office, and the reviewer comments.

In addition to the reviewers' comments I have three of my own:

1) I would like more details about how you reached the concentrations of BDQ and Curcumin that you tested, both for in vitro and in vivo experiments. For example, a dose response curve for each drug individually for some of the experiments in Fig 1 (ie A-G) would help answer this question for the in vitro portion of the results.

2) You mention that Curcumin can induce apoptosis in human macrophages. Therefore, a measurement of toxicity to the host macrophage for experiment Fig 1H is required.

3) It is wholly unclear to me how you came to the speculation that CUR interferes with the electron transport chain. You mention that based on "molecular docking studies, one of CUR's predicted targets is NAD+ dependent DNA ligase". What is the connection between this and disruption of the electron transport chain?

The English in the manuscript needs to be proof-read and revised, many grammatical errors. Please make sure this is done before re-submitting.

Revision Guidelines

Sincerely,
Kayvan Zainabadi
Editor
Microbiology Spectrum

Reviewer #1 (Comments for the Author):

I appreciate the authors' efforts to study the adjunctive effect of Curcumin when added to Bedaquiline to treat *M. abscessus* infection in an in-vitro macrophage model, and an in-vivo mouse model, and to try to establish a mechanism by which Curcumin may enhance the effects of Bedaquiline.

I believe the in-vitro evaluation of the Curcumin-BDQ combination is valid and the first step in studying the additive effect of Curcumin.

I am not sure that the conclusions of 'in a mouse infection model, the combination accelerated bacterial clearance from the lungs' are supported by the reported data.

Figure 1, section I reports CFUs of *M. abscessus* at Day 1 and Day 3, while the paper states mice were sacrificed and organs harvested at Day 3 and Day 7. Is this an error in the figure, or was the Day 7 data not reported? Also, the paper states that 'livers' were harvested and homogenized, while the figure refers to 'lungs'- was this an error? Furthermore, there does not appear to be a statistically significant difference in CFUs between the Bedaquiline group and the combined Curcumin-Bedaquiline group in this figure.

If pathology cross sections of lungs are available, and can be analyzed quantitatively for the presence of infection/lesions, this could help further the claims of the combined treatment being more effective.

Notably, it appears that the CFUs of *M. abscessus* decreased over time even in the control group, which would indicate spontaneous clearance of the microbe, rather than a true infection where the CFUs remain stable or increase over time. This calls into question the utility of such a model to study human infection. I would recommend that this in-vivo model be repeated in mice with a longer-term model of infection in C57Bl/6 mice or a novel immunocompromised mouse strain in which persistent or increasing infection can be demonstrated. Other researchers have utilized corticosteroid treatment in mice to sustain an active, proliferating infection as well as *M. abscessus*-embedded agar beads.

Reviewer #2 (Comments for the Author):

Dear Authors,

To complete the peer review some vital aspects are missing in your methodology section. Please provide a detailed description of how you prepared the curcumin and bedaquiline samples for all experiments. Since both compounds are poor-water soluble the vehicles are extremely important for the results interpretation.

The in vivo study is poorly described methodologically.

**Title: Curcumin Enhances Bedaquiline's Efficacy Against *Mycobacterium***
***abscessus*: In Vitro and In Vivo Evidence**

**Author list**

Dan Luo^{1,2#}, Weile Xie^{1,2#}, Zhe Wang^{1,2*}

**Author affiliations and footnotes**

1. Shanghai Key Laboratory of Veterinary Biotechnology, School of Agriculture and
Biology, Shanghai Jiao Tong University, Shanghai 200240, China

2. Collaborative Innovation Center of Agri-Seeds / School of Agriculture and Biology,
Shanghai Jiao Tong University, Shanghai 200240, China.

#These authors contribute equally to this work.

*Corresponding author: Zhe Wang: wangz@sjtu.edu.cn (corresponding).

**Abstract**

*Mycobacterium abscessus* is a rapidly growing mycobacterium frequently isolated
from clinical samples, and is responsible for severe respiratory, skin, and mucosal
infections in humans. Means to potentiate the effect of existing drugs are urgently

needed. Here we describe the combined effect of the important drug Bedaquiline
(BDQ) and the natural product Curcumin (CUR) on *M. abscessus*. *In vitro*, CUR
enhanced BDQ's inhibitory effect. The combination reduced *M. abscessus* survival
under nutrient-deprived, hypoxic, and acidic conditions, accelerated ATP depletion,
and mitigated the respiratory compensation induced by BDQ. Metabolomics showed
that addition of CUR to BDQ exacerbated the BDQ-dependent downregulation of
purine and pyrimidine metabolism and amino acid synthesis. In RAW264.7

macrophages, the combination reduced bacterial survival more effectively than
monotherapy with either substance alone. Similarly, in a mouse infection model, the
combination accelerated bacterial clearance from the lungs. These findings highlight
the potential of BDQ-CUR combination therapy against *M. abscessus* infections.

**Importance**

*Mycobacterium abscessus* is an emergent pathogen, mainly causes pulmonary
infections, especially in immunocompromised patients. *Mycobacterium abscessus*
shows natural drug resistances to many anti-TB drugs, make itself a nightmare for
infected patients and physicians, therefore raising the need for innovative drug
discovery. In this research, we describe the combined effect of the important drug
Bedaquiline (BDQ) and the natural product Curcumin (CUR) on *M. abscessus*. The *in*
*vitro* and *in vivo* evidence show that CUR enhances BDQ's inhibitory effect. We next
investigated the synergistic effect from metabolism level. The overall findings
highlight the potential of BDQ-CUR combination therapy against *M. abscessus*
infections.

**Keywords**

*Mycobacterium abscessus*, bedaquiline, curcumin, synergy, combination

**Introduction**

Non-tuberculous mycobacteria (NTM) cause chronic, opportunistic pulmonary
infections in susceptible populations[1]. *Mycobacterium abscessus*(*M. abscessus*) is a
prevalent pathogen in NTM[2], [3]. Treatment options are limited due to its natural
resistance to many commonly used antimicrobials. Current treatment, involving 18–

24 months of at least three drugs[4], yields poor outcomes; only 50% of *M. abscessus*
patients and 70% of *Mycobacterium avium complex*(MAC) patients achieve
prolonged culture conversion[5]. Therefore, developing new drugs to treat NTM
infections is urgently needed.

Bedaquiline (BDQ) is an antibiotic approved for treating multidrug-resistant
tuberculosis (MDR-TB). It inhibits the proton pump of mycobacterial ATP synthase,
leading to ATP depletion, unstable pH homeostasis and cell death[6], [7]. BDQ
exhibits moderate to high in vitro activity against NTM[8], [9], [10] and demonstrates
bacteriostatic effects in monotherapy in mouse models of *Mycobacterium avium* (*M.*
*avium*) and *M. abscessus* infections[11], [12]. According to Philley et al., BDQ has
shown potential therapeutic benefits in patients with severe MAC and *M. abscessus*
lung disease, marking it as a promising treatment option for these infections[13].
Although adding BDQ to failing regimens for MAC and *M. abscessus* infections
improved symptoms, it not prevent microbiological failure or the emergence of BDQ
resistance[14], [15]. Therefore, it is necessary to combine BDQ with other drugs to
enhance efficacy and reduce the dosage of antibacterial agents.

Drug repurposing can accelerate drug discovery. Curcumin (CUR) the principal
curcuminoid derived from the plant *Curcuma longa*, has been extensively studied for
its biological and chemical properties. CUR exhibits a broad spectrum of
pharmacological properties, including antimicrobial, anti-inflammatory, antioxidant,
and antitumor activities. Pharmacological studies suggest that CUR has significant
protective effects against TB[16]. Pankul et al. explored how plant-based natural

products affect the pharmacokinetics of BDQ and found that CUR increased the
plasma concentration of BDQ when combined. [17]. This increase may be due to
CUR's ability to enhance BDQ absorption and slow its metabolism by inhibiting
P-GP-mediated efflux. However, data on the BDQ-CUR combination's effectiveness
against *M. abscessus* are limited, especially *in vivo*. The mechanism behind
BDQ-CUR's antibacterial activity remains largely unknown. In this study, we
evaluated the BDQ-CUR drug combination against *M.abscessus* both *in vitro* and *in*
*vivo*, investigating the underlying molecular mechanisms. This work could uncover a
new treatment approach for *M. abscessus*.

**Results and Discussions**

**Curcumin Enhances Bedaquiline's Efficacy Against *M.abscessus* in vitro**

Time-kill assays were conducted using the *M. abscessus* ATCC19977 strain. As
shown in Fig. 1A, the combination of CUR and BDQ demonstrated sustained
bacteriostatic effects. CUR monotherapy did not exhibit any bactericidal effects.
Although BDQ monotherapy (4 µg/ml) initially inhibited bacterial growth, regrowth
was observed within 7 days. These results demonstrate CUR's ability to effectively
enhance BDQ's *in vitro* growth inhibition against *M. abscessus*. This expands on
previous research, showing that CUR is not only a potential antibiotic resistance
breaker but also an effective adjuvant therapy for BDQ[18].

***In vitro* evaluation of BDQ-CUR combination**

Similar to *Mycobacterium tuberculosis*(*M.tuberculosis*), *M.abscessus* evades
destruction by macrophages and neutrophils post-colonization, resulting in granuloma

formation and survival under harsh conditions such as acidic environments, nutrient
deprivation, and hypoxia[19] [20], [21]. Previous studies have shown that the
sensitivity of *M.tuberculosis* to new candidate drugs varies depending on the
physiological state of the cells (active or inactive) [22], [23]. Specifically, this
includes the non-replicating physiological state of *M.abscessus* under oxygen- and
nutrient-deficient (PBS or single-nutrient) conditions[24], and its cellular state under
acidic conditions [25]. Accurately replicating these *in vivo* conditions *in vitro* is
crucial for studying the infection process and evaluating bactericidal activity. Studies
indicate that under nutrient deprivation, BDQ exhibits bactericidal effects against *M.*
*abscessus*[26]. We assessed the bactericidal effect of the BDQ-CUR combination on
*M. abscessus* by simulating a non-replicative state using a starvation model. The
results showed that the BDQ-CUR combination significantly reduced the survival rate
of *M. abscessus* compared to BDQ alone, with a CFU/ml decrease of 0.8 log₁₀ (Fig.
1B). Additionally, we used the Wayne model and pH range from 6.0 to 4.5 (decreasing
by 0.5 increments) to evaluate the survival of *M. abscessus* under hypoxic and acidic
conditions. Under hypoxia conditions, the BDQ-CUR combination effectively
reduced the survival of *M. abscessus*, with a CFU/ml decrease of 0.75 log₁₀(Fig. 1C).
We discovered that *M. abscessus* has high acid tolerance; although BDQ alone had
some inhibitory effect, the combination with CUR was more effective, reducing
growth by 0.8-1.25 log₁₀(Fig. 1D-G). The ability of the BDQ-CUR combination to
maintain its bactericidal activity under above extreme conditions is particularly
important and may improve therapeutic strategies against prolonged *M. abscessus*

infections.

*M. abscessus* can resist intracellular destruction and establish infections,
prompting us to evaluate the antibacterial activity of the BDQ-CUR combination in
RAW264.7 macrophages. Within three days, we observed that the CFU value of the
BDQ-CUR combination in macrophages was reduced by at least 1 log₁₀ compared to
BDQ or CUR used alone (Fig. 1H). Although CUR lacked in vitro inhibitory or
bactericidal effects against *M. abscessus*, it exhibited superior antibacterial activity in
macrophages compared to BDQ at an early stage. This is consistent with Bai et al.'s
findings in an in vitro human macrophage infection model, where CUR not only
induced autophagy and apoptosis but also activated NF-κB, accelerating the clearance
of *M. tuberculosis*[27]. However, in the absence of macrophages, 50 μM CUR had no
impact on *M. tuberculosis* growth [28], [29], [30]. So, identifying drugs that exert
antibacterial activity by modulating the host immune response is crucial. This strategy
reduces bacterial survival pressure, potentially delaying resistance development to
single drugs or combination therapies.

**In vivo evaluation of BDQ-CUR combination**

Although the anti-*M. abscessus* effects of CUR and the CUR-BDQ combination
in macrophages are promising, their protective effects in host organisms have not yet
been confirmed in animal models. As shown in Fig.1I, the BDQ-CUR combination
significantly reduced the bacterial load in the lungs of mice by day 7 of treatment.
Compared to untreated groups, the BDQ-CUR (6/6), BDQ (3/6), and CUR (2/6)
treatments reduced CFU counts by more than an order of magnitude. Combination

therapy can mitigate lung damage and drug toxicity from bacterial infections and may
prevent the need to increase BDQ doses to combat drug-resistant *M. abscessus* [31],
[32]. Dose-escalating studies have indicated the safety of curcumin at doses as high as
12 g/day over 3 months for human[33]. Previous results have shown that treatment
with 16 or 32 $\mu\text{g/mL}$ of CUR reduced the bacillary lung burden and improved survival
rates in mice infected with drug-sensitive *M.tuberculosis* H37Rv[34], [35]. Our results
further confirmed that CUR is a potential antimycobacterial drug not only against
*M.tuberculosis* but also against *M. abscessus*. CUR may inhibit bacterial intracellular
growth and promote the clearance of drug-sensitive strains by inducing
caspase-3-dependent apoptosis and autophagy. This has been demonstrated in
differentiated THP-1 human monocytes, primary human alveolar macrophages, and
Raw 264.7 cells infected with *M.tuberculosis* H37Rv or MDR clinical isolates[36],
[37]. In addition, CUR may directly affect mycobacterial metabolic pathways, which
are crucial for mycobacterial pathogenicity and host persistence[38]. In summary, we
have demonstrated the efficacy of CUR in treating pulmonary infections in mice,
highlighting its potential as an adjunctive agent to bedaquiline (BDQ).

**Influence of BDQ-CUR on *M. abscessus* ATP flux and respiration**

Given that BDQ exerts its antibacterial effects by targeting the ATP synthase c
subunit and blocking ATP synthesis, we investigated the influence of BDQ alone and
in combination with CUR on ATP flux in *M. abscessus*. We found that BDQ alone
reduces ATP flux in *M. abscessus*, with the most significant depletion observed in the
BDQ-CUR combination, demonstrating dose-dependent effects (Fig.2A). According

to molecular docking studies, one of CUR's predicted targets is NAD⁺ dependent
DNA ligase[16]. We speculate that CUR may interfere with the electron transport
chain (ETC), a disturbance that yields surprising results: CUR enhances the ATP
depletion caused by BDQ in mycobacteria.

To clarify the direct effects of BDQ and CUR, both alone and in combination, we
employed extracellular flux (XF) analysis to measure the oxygen consumption rate
(OCR) of *M. abscessus* in real time, as a marker of oxidative phosphorylation
(OXPHOS) (Fig.2B). We observed that the OCR of BDQ-treated *M. abscessus*
increased in a dose-dependent manner. Dirk A Lamprecht et al. reported that the
increase in respiration induced by BDQ in *M. tuberculosis* is a specific response to
ATP depletion, an attempt to restore energy homeostasis[39]. Given that CUR may
inhibit ETC electron flux, we anticipated a reduction in respiration. As expected, the
OCR of CUR-treated cells decreased in a dose-dependent manner (Fig.2C-D).
Therefore, we investigated whether CUR could inhibit the feedback compensation
mechanism induced by BDQ when combined. Our findings indicate that the
BDQ-CUR combination significantly reduced OCR. This suggests that the BDQ-CUR
combination effectively inhibits *M. abscessus* by suppressing oxidative
phosphorylation, thereby abolishing ATP production (Fig.2E-F).

**Global metabolic response of *M. abscessus* to BDQ-CUR**

To elucidate the downstream effects of ETC perturbation by BDQ, CUR, or their
combination, we employed a metabolomics approach. Using the UPLC-Q-TOF/MS
system, we analyzed the perturbations in the metabolite spectrum of *M. abscessus*,

considering the intricate interplay of multiple feedback loops and the flexibility of the
ETC, which may result in complex and unexpected responses. PLSDA model showed
that all replicates in each group were grouped into the same cluster, CUR group and
untreated group (NC) clear aggregation behavior, while the BDQ-CUR(BC) group
was more similar to the BDQ group (Fig.3A). Among them, 37 metabolites were
significantly down-regulated after CUR treatment, 80 metabolites after BDQ
treatment, and 97 metabolites after BC treatment (Fig.3B). There were shared on 37
differentially expressed metabolites (DEMs) among the three groups, which they
mainly participate in purine and pyrimidine metabolism (Fig.3C-D). This observation
is consistent with previous studies purine and pyrimidine biosynthesis were the most
recurrently affected metabolites among the different drug treatments, which is
important role in early drug produced stress response. In addition, KEGG enrichment
analysis showed that the DEMs in BC and BDQ groups were mainly involved in the
carbohydrate metabolism and energy metabolism, such as citrate cycle (TCA cycle),
pentose phosphate, propanoate pathway and oxidative phosphorylation (Fig.3D).

The trends in DEMs between BDQ-CUR (BC) and BDQ alone are consistent and
more pronounced. We observed that dGDP, ADP, UDP-glucose, alanine, guanosine,
and glutamine were downregulated by more than fourfold in the BC group compared
to the BDQ group (Fig.3E). The substantial downregulation of these key metabolic
intermediates disrupts nucleotide, carbohydrate, amino acid, and energy metabolism
pathways (Fig.3F-G). Our previous research identified glutamine synthetase as a
metabolic vulnerability in *M. tuberculosis* to BDQ, with a strong correlation between

glutamine levels and ATP[40]. Furthermore, we confirmed that *M. abscessus* treated
with the BDQ-CUR combination exhibits a low metabolic state, reduced respiration
rate, and halted macromolecule synthesis (Fig.4). This is consistent with other reports
indicating that MTB remodels its metabolism to compensate for reduced ATP levels,
facilitating redox balance [41] [42], [43], [44], [45]. Jared S. Mackenzie et al. found
that BDQ reprograms *M. tuberculosis* metabolism, rendering it vulnerable to the
genetic disruption of glycolysis and gluconeogenesis, leading to rapid sterilization
when combined with OXPHOS inhibition. This may explain why the combination of
BDQ and CUR effectively inhibits the growth of metabolically fragile strains of *M.*
*abscessus*[46]. An intriguing aspect of these findings is that the inhibition of ATP
synthesis by BDQ leads to the differential inhibition of various ATP-dependent
metabolic processes. This "extension mechanism" aligns with emerging concepts that
modifying the primary targets of antibiotics triggers a cascade of events in bacteria,
ultimately disrupting cellular processes[47]. Furthermore, we have demonstrated that
the combined use of CUR and BDQ can enhance this effect.

**Conclusion**

In summary, our research indicates that the combination of BDQ and CUR not
only inhibits the growth of *M. abscessus* but also exhibit potential therapeutic effects.
These include killing dormant bacteria, enhancing bacterial clearance from the body,
reducing bacterial ATP synthesis and cell respiration, and causing significant
metabolic reprogramming and the cessation of macromolecule synthesis. Although
these results are promising, further clinical studies are essential to validate the safety

and efficacy of this combination therapy in human patients. This study provides a
crucial theoretical foundation for developing more effective treatment strategies for
*M.abscessus* infection.

**Materials and Methods**

**Time-Kill Curve Assays**

Time-kill curve assays were conducted using *M.abscessus* ATCC1997. An
overnight culture of ATCC19977 was diluted in 10 ml of Middlebrook 7H9 broth to
reach a final concentration of approximately 10^7 CFU/ml. Bedaquiline (BDQ) at 4
$\mu\text{g/ml}$ or 1 $\mu\text{g/ml}$ and curcumin (CUR) at the lowest synergistic concentrations were
then added, either individually or in combination. Viable cell counts were determined
at 0, 1, 3-, 7-, 14-, and 28-days post-incubation at 37°C. Counts were performed by
plating 20- μl serial dilutions onto 7H10 agar plates in triplicate. Percent survival was
calculated by dividing the CFU/mL at each time point by the initial CFU/mL,
enabling analysis of the antibacterial effects over time.

**Evaluation of Drug Combination Effects in Hypoxic Dormant *M. abscessus***

We established hypoxic dormancy following the modified Wayne's model
described by Gopinath et al. [48]. To simulate non-replicative conditions under acidity
and nutrient scarcity, *M.abscessus* ATCC19977, grown to late log phase, was washed
twice with PBS and resuspended in 7H9 media at pH levels of 6.0, 5.5, 5.0, and 4.5.
The suspensions were transferred to flasks or microtitre plates. BDQ at 0.5 $\mu\text{g/ml}$ or
0.25 $\mu\text{g/ml}$ and CUR at its lowest synergistic concentration were added either singly

or in combination, and incubated at 37°C. Viable counts were determined at 0, 1, 3, 5,
7, and 10 days by plating 20- μ l serial dilutions onto 7H10 agar plates in triplicate.
Survival rates were calculated by the ratio of CFU/mL at each time point to the initial
CFU/mL.

**Cellular ATP Level Measurement in *M. abscessus***

*M.abscessus* ATCC19977 was cultured in 7H9 broth until reaching mid-log phase.
The bacteria were then diluted to approximately 10^6 CFU/ml. Treatments were
administered with BDQ at 10 μ g/ml, CUR at 500 μ g/ml or 200 μ g/ml, or a
combination of both, and incubated for 4 hours at 37°C. The assay was performed
according to the instructions provided with the ATP assay kit by Beyotime.
Luminescence was measured using an illuminometer with a 10-second integration
time per well. Results were analyzed and plotted using GraphPad Prism 9 software.

**Measurement of Oxygen Consumption Rate in *M. abscessus***

Overnight-cultured *M.abscessus* cells were centrifuged and washed twice in
minimal medium devoid of carbon sources. The cells were then diluted to an OD600
of 0.0015 using the same carbon-free 7H9 medium. A volume of 180 μ L of the diluted
cell suspension was added to XF cell culture microplates, which had been pre-coated
with poly-D-lysine to ensure cell adhesion. Prior to antibiotic treatment, the initial
oxygen consumption rate (OCR) was measured for three cycles of four minutes each
to establish a baseline. These measurements were conducted using a Seahorse XFe96
Analyzer (Agilent, USA), as per the manufacturer's instructions. For the metabolic
assay, glucose was automatically injected into each well at a concentration of 2

265 mg/mL via the A ports of the sensor cartridge. Subsequently, treatments with BDQ,
CUR, and carbon-free 7H9 medium were administered through the remaining ports.
The OCR was then measured for four cycles of seven minutes each following each
antibiotic injection.

**LC-MS-Based Metabolomics Analysis of *M. abscessus***

Metabolomic profiling of *M.tuberculosis* was performed in line with
methodologies described in prior literature[46]. Metabolites were separated using a
Vanquish UHPLC system, which was integrated with a Q Exactive Plus Mass
Spectrometer, both from Thermo Fisher Scientific, USA. Metabolite identification
was facilitated by TraceFinder software (Thermo Fisher Scientific, USA). Statistical
analysis was carried out to identify differentially expressed metabolites (DEMs),
defined by fold changes of at least 1.5 and p-values of 0.05 or less. Data visualization
and statistical tests were executed using R software, version 3.6.3. The primary liquid
chromatography-mass spectrometry data have been deposited in the EMBL-EBI
MetaboLights database under the identifier MTBLS10177.

**Evaluation of Drug Combination Treatment Effects of *M. abscessus* Infection in**

**RAW 264.7 Macrophages Infections**

RAW 264.7 macrophages were seeded in 6-well culture plates at a density of $1 \times$
10^6 cells per well. To prepare for infection, the cells were washed three times with
DMEM to remove nonadherent cells. The macrophages were then infected with
*M.abscessus* at a multiplicity of infection (MOI) of 1 for 4 hours. Following the
infection, the cells were washed twice with prewarmed 1x PBS and treated with

DMEM medium containing 500 µg/mL kanamycin to eliminate non-internalized
bacteria. Drug treatment was by adding BDQ at 1 µg/ml and CUR at 500 µg/ml, either
individually or in combination, to the infected cells. The cells were then incubated for
additional periods of 24 and 72 hours. At these specified time points, the cells were
lysed using sterile 0.9% saline containing 0.025% SDS. The resulting lysates were
serially diluted and plated on 7H10 agar plates in triplicate. CFU were calculated to
determine the bacterial load.

**In Vivo Treatment Evaluation**

Female C57BL/6 mice, 12 per group, were anaesthetized and then inoculated
nasally with *M. abscessus* at a concentration of 10⁹ CFU. One day post-inoculation,
treatments began with BDQ at 30mg / kg and CUR at 200 mg/kg, administered orally
either singly or in combination. The treatments were given once every other day. A
control group received the same volume of solvent. Five or six mice from each group
were sacrificed on days 3 and 7 post-infection. The livers were aseptically excised to
observe gross lesions, then homogenized and plated on 7H10 agar plates for CFU
counts. All procedures were conducted in accordance with the Guidelines of the
Animal Care and Use Committee of Shanghai Jiao Tong University. The animal study
protocols were approved by the Institutional Animal Care and Use Committee of
Shanghai Jiao Tong University.

**Statistical Analysis**

All data were presented as the mean ± SD from a minimum of three independent
experiments. Statistical analyses were conducted using GraphPad Prism software

version 8.0.1. Differences were considered statistically significant at $p < 0.05$ (*), $p <$
0.01 (**), $p < 0.001$ (***) and $p < 0.0001$ (****). The notation 'ns' indicates a lack of
statistical significance

**Funding**

This research was kindly supported by a grant from the National Key Research
and Development Plans of China (No. 2021YFD1800401) to Zhe Wang, National
Natural Science Foundation of China (No. 32070128) to Zhe Wang, and Shanghai
Biomedical Science and Technology Support Special Project (No. 21S11900200) to
Zhe Wang.

**Acknowledgement**

We thank Prof. Kyu Y. Rhee (Weill Cornell Medical College) and all members
of Wang lab for critical discussion and document revision.

**Declaration of interests**

The authors declare that they have no competing interests.

**References**

- [1] D. E. Griffith *et al.*, “An official ATS/IDSA statement: diagnosis, treatment, and
prevention of nontuberculous mycobacterial diseases,” *Am. J. Respir. Crit. Care*
*Med.*, vol. 175, no. 4, pp. 367–416, Feb. 2007, doi:
10.1164/rccm.200604-571ST.
- [2] W. Hoefsloot *et al.*, “The geographic diversity of nontuberculous mycobacteria
isolated from pulmonary samples: an NTM-NET collaborative study,” *Eur.*
*Respir. J.*, vol. 42, no. 6, pp. 1604–1613, Dec. 2013, doi:
10.1183/09031936.00149212.
- [3] J. E. Stout, W.-J. Koh, and W. W. Yew, “Update on pulmonary disease due to
non-tuberculous mycobacteria,” *Int. J. Infect. Dis. IJID Off. Publ. Int. Soc. Infect.*
*Dis.*, vol. 45, pp. 123–134, Apr. 2016, doi: 10.1016/j.ijid.2016.03.006.
- [4] R. A. Floto *et al.*, “US Cystic Fibrosis Foundation and European Cystic Fibrosis
Society consensus recommendations for the management of non-tuberculous
mycobacteria in individuals with cystic fibrosis: executive summary,” *Thorax*,
338 vol. 71, no. 1, pp. 88–90, Jan. 2016, doi: 10.1136/thoraxjnl-2015-207983.
- [5] J. van Ingen, B. E. Ferro, W. Hoefsloot, M. J. Boeree, and D. van Soolingen,

- “Drug treatment of pulmonary nontuberculous mycobacterial disease in
HIV-negative patients: the evidence,” *Expert Rev. Anti Infect. Ther.*, vol. 11, no.
10, pp. 1065–1077, Oct. 2013, doi: 10.1586/14787210.2013.830413.
- [6] M. Rao, T. L. Streur, F. E. Aldwell, and G. M. Cook, “Intracellular pH regulation
by *Mycobacterium smegmatis* and *Mycobacterium bovis* BCG,” *Microbiol.*
*Read. Engl.*, vol. 147, no. Pt 4, pp. 1017–1024, Apr. 2001, doi:
10.1099/00221287-147-4-1017.
- [7] K. Andries *et al.*, “A diarylquinoline drug active on the ATP synthase of
*Mycobacterium tuberculosis*,” *Science*, vol. 307, no. 5707, pp. 223–227, Jan.
2005, doi: 10.1126/science.1106753.
- [8] B. A. Brown-Elliott, J. V. Philley, D. E. Griffith, F. Thakkar, and R. J. Wallace,
“In Vitro Susceptibility Testing of Bedaquiline against *Mycobacterium avium*
Complex,” *Antimicrob. Agents Chemother.*, vol. 61, no. 2, pp. e01798-16, Feb.
2017, doi: 10.1128/AAC.01798-16.
- [9] E. Huitric, P. Verhasselt, K. Andries, and S. E. Hoffner, “In vitro
antimycobacterial spectrum of a diarylquinoline ATP synthase inhibitor,”
*Antimicrob. Agents Chemother.*, vol. 51, no. 11, pp. 4202–4204, Nov. 2007, doi:
10.1128/AAC.00181-07.
- [10] B. Li *et al.*, “Determination of MIC Distribution and Mechanisms of Decreased
Susceptibility to Bedaquiline among Clinical Isolates of *Mycobacterium*
*abscessus*,” *Antimicrob. Agents Chemother.*, vol. 62, no. 7, pp. e00175-18, Jul.
2018, doi: 10.1128/AAC.00175-18.
- [11] I. Lerat *et al.*, “In vivo evaluation of antibiotic activity against *Mycobacterium*
*abscessus*,” *J. Infect. Dis.*, vol. 209, no. 6, pp. 905–912, Mar. 2014, doi:
10.1093/infdis/jit614.
- [12] N. Lounis, T. Gevers, J. Van den Berg, L. Vranckx, and K. Andries, “ATP
synthase inhibition of *Mycobacterium avium* is not bactericidal,” *Antimicrob.*
*Agents Chemother.*, vol. 53, no. 11, pp. 4927–4929, Nov. 2009, doi:
10.1128/AAC.00689-09.
- [13] J. V. Philley *et al.*, “Preliminary Results of Bedaquiline as Salvage Therapy for
Patients With Nontuberculous Mycobacterial Lung Disease,” *Chest*, vol. 148, no.
2, pp. 499–506, Aug. 2015, doi: 10.1378/chest.14-2764.
- [14] J. V. Philley *et al.*, “Preliminary Results of Bedaquiline as Salvage Therapy for
Patients With Nontuberculous Mycobacterial Lung Disease,” *Chest*, vol. 148, no.
2, pp. 499–506, Aug. 2015, doi: 10.1378/chest.14-2764.
- [15] D. C. Alexander *et al.*, “Emergence of mmpT5 Variants during Bedaquiline
Treatment of *Mycobacterium intracellulare* Lung Disease,” *J. Clin. Microbiol.*,
377 vol. 55, no. 2, pp. 574–584, Feb. 2017, doi: 10.1128/JCM.02087-16.
- [16] N. Barua and A. K. Buragohain, “Therapeutic Potential of Curcumin as an
Antimycobacterial Agent,” *Biomolecules*, vol. 11, no. 9, p. 1278, Aug. 2021, doi:
10.3390/biom11091278.
- [17] P. Kotwal *et al.*, “Effect of Natural Phenolics on Pharmacokinetic Modulation of
Bedaquiline in Rat to Assess the Likelihood of Potential Food-Drug Interaction,”
*J. Agric. Food Chem.*, vol. 68, no. 5, pp. 1257–1265, Feb. 2020, doi:

- 10.1021/acs.jafc.9b06529.
- [18] E. Marini *et al.*, “Curcumin, an antibiotic resistance breaker against a
multiresistant clinical isolate of *Mycobacterium abscessus*,” *Phytother. Res. PTR*,
387 vol. 32, no. 3, pp. 488–495, Mar. 2018, doi: 10.1002/ptr.5994.
- [19] M. D. Johansen, J.-L. Herrmann, and L. Kremer, “Non-tuberculous
mycobacteria and the rise of *Mycobacterium abscessus*,” *Nat. Rev. Microbiol.*,
390 vol. 18, no. 7, pp. 392–407, Jul. 2020, doi: 10.1038/s41579-020-0331-1.
- [20] M. Rottman *et al.*, “Importance of T cells, gamma interferon, and tumor necrosis
factor in immune control of the rapid grower *Mycobacterium abscessus* in
C57BL/6 mice,” *Infect. Immun.*, vol. 75, no. 12, pp. 5898–5907, Dec. 2007, doi:
10.1128/IAI.00014-07.
- [21] A. Bernut, M. Nguyen-Chi, I. Halloum, J.-L. Herrmann, G. Lutfalla, and L.
Kremer, “*Mycobacterium abscessus*-Induced Granuloma Formation Is Strictly
Dependent on TNF Signaling and Neutrophil Trafficking,” *PLoS Pathog.*, vol.
12, no. 11, p. e1005986, Nov. 2016, doi: 10.1371/journal.ppat.1005986.
- [22] E. D. Pieterman, M. J. Sarink, C. Sala, S. T. Cole, J. E. M. de Steenwinkel, and
H. I. Bax, “Advanced Quantification Methods To Improve the 18b Dormancy
Model for Assessing the Activity of Tuberculosis Drugs In Vitro,” *Antimicrob.*
*Agents Chemother.*, vol. 64, no. 7, pp. e00280-20, Jun. 2020, doi:
10.1128/AAC.00280-20.
- [23] E. G. Salina, O. Ryabova, A. Vocat, B. Nikonenko, S. T. Cole, and V. Makarov,
“New 1-hydroxy-2-thiopyridine derivatives active against both replicating and
dormant *Mycobacterium tuberculosis*,” *J. Infect. Chemother. Off. J. Jpn. Soc.*
*Chemother.*, vol. 23, no. 11, pp. 794–797, Nov. 2017, doi:
10.1016/j.jiac.2017.04.012.
- [24] Y.-K. Yam, N. Alvarez, M.-L. Go, and T. Dick, “Extreme Drug Tolerance of
*Mycobacterium abscessus* ‘Persists,’” *Front. Microbiol.*, vol. 11, p. 359, 2020,
doi: 10.3389/fmicb.2020.00359.
- [25] A. Lanni *et al.*, “Activity of Drug Combinations against *Mycobacterium*
*abscessus* Grown in Aerobic and Hypoxic Conditions,” *Microorganisms*, vol. 10,
no. 7, p. 1421, Jul. 2022, doi: 10.3390/microorganisms10071421.
- [26] A. L. Mulyukin *et al.*, “Distinct Effects of Moxifloxacin and Bedaquiline on
Growing and ‘Non-Culturable’ *Mycobacterium abscessus*,” *Microorganisms*, vol.
11, no. 11, p. 2690, Nov. 2023, doi: 10.3390/microorganisms11112690.
- [27] X. Bai *et al.*, “Curcumin enhances human macrophage control of
*Mycobacterium tuberculosis* infection,” *Respirol. Carlton Vic*, vol. 21, no. 5, pp.
951–957, Jul. 2016, doi: 10.1111/resp.12762.
- [28] M. Shariq *et al.*, “*Mycobacterium tuberculosis* RipA Dampens TLR4-Mediated
Host Protective Response Using a Multi-Pronged Approach Involving
Autophagy, Apoptosis, Metabolic Repurposing, and Immune Modulation,” *Front.*
*Immunol.*, vol. 12, p. 636644, Mar. 2021, doi: 10.3389/fimmu.2021.636644.
- [29] E. Arnett *et al.*, “PPAR γ is critical for *Mycobacterium tuberculosis* induction of
Mcl-1 and limitation of human macrophage apoptosis,” *PLoS Pathog.*, vol. 14,
no. 6, p. e1007100, Jun. 2018, doi: 10.1371/journal.ppat.1007100.

- [30] A. Bah, M. Sanicas, J. Nigou, C. Guilhot, C. Astarie-Dequeker, and I. Vergne,
“The Lipid Virulence Factors of Mycobacterium tuberculosis Exert Multilayered
Control over Autophagy-Related Pathways in Infected Human Macrophages,”
*Cells*, vol. 9, no. 3, p. 666, Mar. 2020, doi: 10.3390/cells9030666.
- [31] T. Gao *et al.*, “Antimicrobial Effect of Oxazolidinones and Its Synergistic Effect
with Bedaquiline Against Mycobacterium abscessus Complex,” *Infect. Drug*
*Resist.*, vol. 16, pp. 279–287, 2023, doi: 10.2147/IDR.S395750.
- [32] A. Lanni *et al.*, “Activity of Drug Combinations against Mycobacterium
abscessus Grown in Aerobic and Hypoxic Conditions,” *Microorganisms*, vol. 10,
no. 7, p. 1421, Jul. 2022, doi: 10.3390/microorganisms10071421.
- [33] S. C. Gupta, S. Patchva, and B. B. Aggarwal, “Therapeutic Roles of Curcumin:
Lessons Learned from Clinical Trials,” *AAPS J.*, vol. 15, no. 1, pp. 195–218,
Nov. 2012, doi: 10.1208/s12248-012-9432-8.
- [34] J. V. Lara-Espinosa *et al.*, “Effect of Curcumin in Experimental Pulmonary
Tuberculosis: Antimycobacterial Activity in the Lungs and Anti-Inflammatory
Effect in the Brain,” *Int. J. Mol. Sci.*, vol. 23, no. 4, p. 1964, Feb. 2022, doi:
10.3390/ijms23041964.
- [35] S. A. Marathe, I. Dasgupta, D. P. Gnanadhas, and D. Chakravorty, “Multifaceted
roles of curcumin: two sides of a coin!,” *Expert Opin. Biol. Ther.*, vol. 11, no. 11,
pp. 1485–1499, Nov. 2011, doi: 10.1517/14712598.2011.623124.
- [36] X. Bai *et al.*, “Curcumin enhances human macrophage control of
Mycobacterium tuberculosis infection,” *Respirol. Carlton Vic*, vol. 21, no. 5, pp.
951–957, Jul. 2016, doi: 10.1111/resp.12762.
- [37] P. K. Gupta, S. Kulkarni, and R. Rajan, “Inhibition of Intracellular Survival of
Multi Drug Resistant Clinical Isolates of Mycobacterium tuberculosis in
Macrophages by Curcumin,” *Open Antimicrob. Agents J.*, vol. 4, no. 1, Nov.
2013, Accessed: Jun. 06, 2024. [Online]. Available:
<https://benthamopen.com/ABSTRACT/TOANTIMJ-4-1>
- [38] A. K. Singh *et al.*, “Identification of lipid metabolism-targeting compounds
active against drug-resistant M. tuberculosis,” *J. Glob. Antimicrob. Resist.*, vol. 7,
pp. 26–27, Dec. 2016, doi: 10.1016/j.jgar.2016.07.003.
- [39] D. A. Lamprecht *et al.*, “Turning the respiratory flexibility of Mycobacterium
tuberculosis against itself,” *Nat. Commun.*, vol. 7, no. 1, Art. no. 1, Aug. 2016,
doi: 10.1038/ncomms12393.
- [40] Z. Wang *et al.*, “Mode-of-action profiling reveals glutamine synthetase as a
collateral metabolic vulnerability of M. tuberculosis to bedaquiline,” *Proc. Natl.*
*Acad. Sci. U. S. A.*, vol. 116, no. 39, pp. 19646–19651, Sep. 2019, doi:
10.1073/pnas.1907946116.
- [41] A. Koul *et al.*, “Delayed bactericidal response of Mycobacterium tuberculosis to
bedaquiline involves remodelling of bacterial metabolism,” *Nat. Commun.*, vol.
5, p. 3369, Feb. 2014, doi: 10.1038/ncomms4369.
- [42] T. Q. Nguyen *et al.*, “Synergistic Effect of Q203 Combined with PBTZ169
against Mycobacterium tuberculosis,” *Antimicrob. Agents Chemother.*, vol. 66,
no. 12, p. e0044822, Dec. 2022, doi: 10.1128/aac.00448-22.

- [43] S. Kim *et al.*, “Evaluating the effect of clofazimine against Mycobacterium
tuberculosis given alone or in combination with pretomanid, bedaquiline or
linezolid,” *Int. J. Antimicrob. Agents*, vol. 59, no. 2, p. 106509, Feb. 2022, doi:
10.1016/j.ijantimicag.2021.106509.
- [44] M. Lindman and T. Dick, “Bedaquiline Eliminates Bactericidal Activity of
β -Lactams against Mycobacterium abscessus,” *Antimicrob. Agents Chemother.*,
478 vol. 63, no. 8, pp. e00827-19, Jul. 2019, doi: 10.1128/AAC.00827-19.
- [45] M. M. Ruth *et al.*, “A bedaquiline/clofazimine combination regimen might add
activity to the treatment of clinically relevant non-tuberculous mycobacteria,” *J.*
*Antimicrob. Chemother.*, vol. 74, no. 4, pp. 935–943, Apr. 2019, doi:
10.1093/jac/dky526.
- [46] J. S. Mackenzie *et al.*, “Bedaquiline reprograms central metabolism to reveal
glycolytic vulnerability in Mycobacterium tuberculosis,” *Nat. Commun.*, vol. 11,
no. 1, p. 6092, Nov. 2020, doi: 10.1038/s41467-020-19959-4.
- [47] T. Dick and D. Young, “How antibacterials really work: impact on drug
discovery,” *Future Microbiol.*, vol. 6, no. 6, pp. 603–604, Jun. 2011, doi:
10.2217/fmb.11.26.
- [48] S. Raghunandan, L. Jose, and R. A. Kumar, “Dormant Mycobacterium
tuberculosis converts isoniazid to the active drug in a Wayne’s model of
dormancy,” *J. Antibiot. (Tokyo)*, vol. 71, no. 11, pp. 939–949, Nov. 2018, doi:
10.1038/s41429-018-0098-z.

**Figure legends**

**Fig.1. Curcumin Enhances Bedaquiline's Efficacy Against *M. abscessus* in vitro**
**and in vivo.** A. Time Kill Kinetics: Growth curves of *M. abscessus* in 7H9 medium
treated with varying concentrations of BDQ and CUR, including control and
single-drug treatments, over a period of 28 days. B. Starvation Model: Survival curves
of *M. abscessus* in nutrient-poor conditions under combined BDQ and CUR treatment
over 10 days. C. Hypoxia State: Survival of *M. abscessus* in hypoxic conditions under
combined BDQ and CUR treatment over 10 days. D-G. Acidic Conditions: Survival
curves of *M. abscessus* at different pH levels (6.0, 5.5, 5.0, 4.5) under combined BDQ
and CUR treatment. H. RAW264.7 Macrophages: Evaluation of BDQ and CUR
combined treatment on *M. abscessus* infection in RAW264.7 macrophages. I. Mouse

Lung Infection Model: Efficacy of BDQ and CUR combined treatment in mice with
*M. abscessus* lung infection.

**Fig. 2. Impact of BDQ and CUR on ATP Production and Cellular Respiration in**

*M. abscessus*. A. ATP Content Analysis: Relative changes in ATP concentration in *M.*

*abscessus* following treatment with BDQ and CUR, administered both individually

and in combination. CUR was applied at high (500 µg/mL) and low (200 µg/mL)

doses, while BDQ was used at 10 times its minimum inhibitory concentration (MIC).

B. Schematic of Extracellular Flux (XF) Assay: Diagram illustrating the principles

and workflow of the XF assay used to measure the oxygen consumption rate (OCR)

and other bioenergetic parameters in *M. abscessus*. Drug compounds were introduced

through ports, with dissolved O₂ and pH monitored continuously. C-D. Bioenergetic

Profiles: OCR of *M. abscessus* when exposed to varying concentrations of BDQ or

CUR. Exposure to BDQ results in a concentration-dependent increase in respiration,

whereas CUR exposure leads to a concentration-dependent decrease. E-F. Effects of

BDQ and CUR Combinations on Respiration: Bioenergetic profiles depicting the

effects of sequential drug administration on OCR. Panel E shows the response when

CUR is added prior to BDQ, and Panel F demonstrates the response when BDQ is

added before CUR. Both sequences effectively reduce cellular respiration compared

to treatment with BDQ alone. Data were analyzed using Seahorse XF Wave software,

with standard deviations calculated from four replicates.

**Fig. 3. Metabolic Responses of *M. abscessus* to BDQ-CUR Combination**

**Treatment**

528 A. PLSDA Analysis: Metabolite profiles in *M. abscessus* treated with BDQ, CUR
alone, and BDQ-CUR combination (BC). The plot demonstrates clustering by
treatment group.

B. Volcano Plots: Differential metabolite expression in *M. abscessus* under treatments
with BDQ alone, CUR alone, and their combination (BC). Significantly different
metabolites (DEMs) are highlighted ($p \leq 0.05$ and fold change ≥ 1.5); blue dots
indicate downregulated metabolites, and red dots indicate upregulated metabolites. C.

Venn Diagram: Overlap of differential metabolites identified in *M. abscessus* under
treatments with BDQ, CUR, and BC. D. KEGG Pathway Enrichment Analysis:

Analysis of differential metabolites under individual and combined BDQ and CUR
treatments, highlighting significant pathways ($FDR \leq 0.001$). E. Fold Changes in Key

Metabolites Abundance: Bar graph depicting \log_2 fold changes of key differential
metabolites under BDQ and CUR treatments compared to control, BDQ alone, and

CUR alone, respectively. F. Network Diagram of Key Metabolites: Visualization of
key differential metabolites, with blue nodes indicating downregulation and node size

reflecting the degree of connectivity. G. KEGG Pathway Enrichment Comparison:

Pathway enrichment analysis for metabolites differentially expressed under the BDQ
and CUR combination treatment compared to single-drug treatments.

**Fig. 4. Impact of BDQ-CUR Combination on the Metabolic State of *M. abscessus***

BDQ primarily targets ATP synthase, enhancing cellular respiration by reducing the
negative regulation of the respiratory chain, a compensatory mechanism to counteract
the drug's effects. In contrast, CUR impedes the conversion of NADH to NAD⁺,
thereby inhibiting cellular respiration. While BDQ alone only inhibits the growth of
*M. abscessus* in vitro at early stages, the BDQ-CUR combination leads to a significant
synergistic effect, extending the efficacy of BDQ. CUR blocks the respiratory
compensation induced by BDQ, thereby enhancing BDQ's antimycobacterial efficacy.
Our study reveals that the CUR-BDQ combination disrupts BDQ-induced
compensation mechanisms, resulting in a substantial decrease in the metabolic
activity of *M. abscessus*. This decrease is characterized by marked reductions in the
ATP-ADP-AMP cycle and the metabolism of central nitrogen sources such as
glutamine and glutamate, leading to a comprehensive downregulation of key
metabolic pathways, including amino acid synthesis, nucleotide synthesis, and central
carbon metabolism. This figure was created using BioRender.com.

**Table S1. Differential Metabolites of *M. abscessus* Treated with BDQ, CUR, and**
**BDQ-CUR Combination.**

Dear Editors and Reviewers:

We sincerely thank the reviewers for your thoughtful and constructive comments, which have greatly improved the quality of our manuscript (**Spectrum 02295-24**). Below, we provide point-by-point responses to address each comment, accompanied by the corresponding revisions in the manuscript.

Point-by-point responses to Editors and Reviewers' comments

Editors:

1) I would like more details about how you reached the concentrations of BDQ and Curcumin that you tested, both for in vitro and in vivo experiments. For example, a dose response curve for each drug individually for some of the experiments in Fig 1 (ie A-G) would help answer this question for the in vitro portion of the results.

Response: Thank you for your insightful comments. In our preliminary experiments, we conducted dose-response studies to determine the effective concentrations of BDQ and Curcumin (CUR). While these data were omitted from our initial submission for brevity, we now provide a detailed explanation below:

CUR dose chooses:

Previous reports indicate that the MIC of CUR against *M. abscessus* (29904) ranges from 128 to 256 $\mu\text{g/mL}$ [1]. Utilizing a resazurin assay, we assessed concentrations of 100, 200, and 500 $\mu\text{g/mL}$. However, the inherent color of CUR interfered with resazurin readings. To address this issue, we evaluated its effects on the growth curve of *M. abscessus* ATCC 19977 (**Fig. RS1**). No significant inhibitory effects were observed at these concentrations. Based on prior findings, we selected 200 $\mu\text{g/mL}$ and 500 $\mu\text{g/mL}$ as experimental conditions for combination studies with BDQ.

Fig. RS1. Growth curve with CUR treatment in *M. abscessus*.

Combination with BDQ:

Subsequently, we investigated the in vitro bactericidal effects of CUR at concentrations of 200 µg/mL and 500 µg/mL in combination with BDQ at 1×MIC, 4×MIC, and 8×MIC over a 28-day period (**Fig. RS2**). The results revealed the following:

- At 8×MIC of BDQ alone, *M. abscessus* exhibited partial regrowth by day 28.
- At 4×MIC of BDQ alone, bacterial growth plateaued by day 28, entering a stationary phase.
- At 1×MIC of BDQ alone, the stationary phase was reached as early as day 7.

Notably, the addition of CUR significantly enhanced BDQ's bactericidal activity across all tested concentrations. Specifically, the combination with CUR at 500 µg/mL extended the time required for bacterial regrowth or stabilization at 1×MIC, 4×MIC, and 8×MIC of BDQ. This synergistic effect highlights CUR's potential to enhance BDQ's efficacy.

In the main text, we included representative curves for the combination of 1×MIC and 4×MIC BDQ with 500 µg/mL CUR to emphasize the combination effect (Fig. 1A).

Fig. RS2. Time-Kill Kinetics of BDQ and CUR combinations

BDQ Sub-MIC Studies

BDQ is an essential component in the treatment regimens for drug-resistant TB in humans; however, its clinical application is constrained by significant toxicity concerns, such as QT prolongation, hepatotoxicity, and phospholipidosis [2], [3], [4], [5], [6]. In contrast, CUR has no reported toxic or side effects in humans, making it an ideal candidate for combination therapies aimed at reducing the necessary dosage of BDQ.

Given BDQ's potential toxicity, we tested sub-MIC levels ($1/4 \times \text{MIC}$ and $1/2 \times \text{MIC}$) in combination with CUR. These lower concentrations also effectively inhibited bacterial growth, demonstrating CUR's potentiating effects (**Fig. RS3**). However, as higher concentrations were sufficient to highlight synergy, sub-MIC data were excluded from the main text for brevity.

Fig. RS3. Time-Kill Kinetics with sub-MIC concentrations of BDQ and CUR, over a period of 6 days.

In vitro Stress Models

BDQ exhibits bactericidal activity against *M. abscessus* under stress conditions such as nutrient deprivation, acidic stress, and hypoxia [7], [8], [9], [10]. Even at inhibitory or sub-inhibitory concentrations, BDQ remains effective in these non-replicating models, highlighting its efficacy in challenging environments.

We tested combinations of 1/4×MIC and 1/2×MIC BDQ with CUR (200 µg/mL and 500 µg/mL). The results showed that CUR significantly potentiated BDQ's activity across all conditions (**Fig. R4**). To emphasize the most relevant findings, we presented the results for 1/4×MIC and 1/2×MIC BDQ with 500 µg/mL CUR in the main text (Fig. 1B-G).

Fig. RS4. Survival curves under acidic, hypoxic, and starvation conditions

Macrophage Infection Model

We evaluated the therapeutic efficacy of BDQ at 1×MIC and 1/2×MIC in combination with CUR at concentrations of 200 µg/mL and 500 µg/mL in macrophage infection models. The findings indicated that CUR significantly enhanced the therapeutic effects of BDQ on infected macrophages (**Fig. RS5**).

To maintain consistency with the data presentation in other sections and emphasize the combination effect, we included the results for the combination of 1×MIC BDQ with 500 µg/mL CUR in the main text (Fig. 1H).

Fig. RS5. Evaluation of BDQ and CUR combined treatment on *M. abscessus* infection in RAW264.7 macrophages.

Summary

These additional data offer a comprehensive rationale for the selected concentrations of BDQ and CUR in our study. We hope this explanation adequately addresses your concerns and appreciate your valuable feedback, which has contributed to the refinement of our manuscript.

2) You mention that Curcumin can induce apoptosis in human macrophages. Therefore, a measurement of toxicity to the host macrophage for experiment Fig 1H is required.

Response: Thank you for raising this important concern. CUR has been identified as a potential anticancer agent capable of inducing apoptosis in various cancer cells [11]. However, studies have shown that its apoptotic effect on THP-1 cells is abolished after PMA-induced differentiation [12]. Additionally, CUR's impact on the viability of RAW264.7 cells has been reported to occur only at concentrations exceeding 100 μ M (\sim 37 μ g/mL) [13].

To address this, we re-evaluated the toxicity of CUR on RAW264.7 cells during our initial experimental design. The results confirmed that CUR at 500 μ g/mL (\sim 1.35 mM) is safe for RAW264.7 cells, and even at 1000 μ g/mL (\sim 2.7 mM), no significant toxicity was observed (**Fig. RS6**). These findings validate the safety of the CUR concentrations used in our experiments.

Fig. RS6. Toxicity evaluation of CUR on RAW264.7.

3) It is wholly unclear to me how you came to the speculation that CUR interferes with the electron transport chain. You mention that based on "molecular docking studies, one of CUR's predicted targets is NAD⁺ dependent DNA ligase". What is the connection between this and disruption of the electron transport chain?

Response: Thank you for your valuable comments. We acknowledge that our previous explanation lacked clarity and logical coherence. Below is the revised response, incorporating additional data to provide a cohesive explanation.

Previous studies have identified that NAD⁺-dependent DNA ligase as a potential target of CUR through molecular docking analyses[14]. This intriguing finding motivated us and prompted further investigation into the effect of CUR on NAD⁺-related metabolic pathways in *M. abscessus*.

In our experiments, CUR treatment led to a significant increase in intracellular NAD⁺ levels (**Fig. RS7**). This unexpected result suggests that CUR may disrupt with NAD⁺-dependent enzymatic processes, potentially including NAD⁺-dependent DNA ligase or other NAD⁺-consuming enzymes. To investigate this further, we examined the combined effect of CUR and BDQ on NAD⁺ metabolism and redox balance.

Fig. RS7. Changes in NAD(H) levels following CUR treatment

Interestingly, when CUR was combined with BDQ, a strikingly different effect was observed: the intracellular NADH/NAD⁺ ratio increased significantly (**Fig. RS8**). This change was characterized by a reduction in NAD⁺ alongside a simultaneous increase in NADH levels, disrupted the NAD⁺/NADH homeostasis. This observation contrasts with the NAD⁺ accumulation seen with CUR alone, suggesting that CUR enhances BDQ's impact on cellular redox balance in a synergistic manner.

Moreover, previous studies have demonstrated a strong correlation between the bactericidal activity of ETC-targeted drug combinations and increased NADH/NAD⁺ ratios[15]. Consistent with these findings, our results showed that the BDQ-CUR combination led to a three-fold increase in the NADH/NAD⁺ ratio compared to BDQ alone. This enhanced redox imbalance was accompanied by a marked reduction in ATP levels, providing a plausible mechanism for the synergistic bactericidal effect of the BDQ-CUR combination.

Fig. RS8. Effects of BDQ-CUR combination on NAD(H) balance

Revised Content in Manuscript

The revised manuscript now includes the following updated results and discussion (**Line172-196**):

Molecular docking studies have identified NAD-dependent DNA ligase as one of the predicted targets of CUR [16]. To explore this interaction, we assessed the effect of CUR on NAD⁺ levels and found that CUR significantly increased intracellular NAD⁺ levels and the NAD⁺/NADH ratio (Fig. 3B). NAD(H) homeostasis plays a crucial role in drug susceptibility and infection processes in mycobacteria [43], [44], [45]. While NAD⁺ depletion can induce lethal low-energy states, excessive NAD⁺ accumulation elevates ROS levels, which can also be toxic. Oxidative stressors like H₂O₂ and HClO sharply increase NAD⁺ levels and the NAD⁺/NADH ratio, leading to lethal effect. Similarly, clofazimine treatment initiates an over-driven ROS cycle pathway, leading to an increase in NAD⁺ and ROS, effectively killing mycobacterium[46], [47], [48], [49]. These challenges are often mitigated by mycobacteria through metabolic adaptations to withstand such stresses [46]. *M. abscessus*, in particular, exhibits greater adaptability to ROS enrichment[50], [51]. This adaptability may explain the lack of CUR-mediated inhibition observed in our study, as CUR-induced alterations in NAD(H) homeostasis appear insufficient to significantly impact its growth. Additionally, previous studies have reported a strong correlation between the bactericidal activity of ETC-targeted drug combinations and an increased intracellular NADH/NAD⁺ ratio[52], [53], [54]. Consistent with this, our results showed that the combination treatment led to a three-fold increase in the NADH/NAD⁺ ratio compared to the BDQ treatment alone. This increase was driven by NADH upregulation in the combination group (Fig. 3C and Fig. 3D). NAD(H) plays a critical role as a respiratory cofactor. Its imbalance initially triggers metabolic adaptation but ultimately leads to redox failure and cellular dysfunction [55], [56], [57]. CUR enhanced BDQ disruption of redox balance and energy metabolism in *M. abscessus*, providing insights into the synergy of this drug combination.

We believe this revised explanation, along with the updated data, better addresses your concerns. Thank you for your insightful feedback, which has greatly improved

the logical flow and clarity of our manuscript.

4)The English in the manuscript needs to be proof-read and revised, many grammatical errors. Please make sure this is done before re-submitting.

Response: Thank you for pointing out this issue. We sincerely value the importance of clear and accurate language in scientific writing. To ensure the highest standard of readability and grammatical correctness, we have had our manuscript professionally edited by Editage, a recognized scientific editing service (<https://www.editage.cn>).

We have also included the certificate of editing as supplementary material to confirm this process. We hope that these efforts will meet your expectations, and we truly appreciate your feedback in helping us improve the quality of our manuscript.

Reviewer 1:

*1) I am not sure that the conclusions of 'in a mouse infection model, the combination accelerated bacterial clearance from the lungs' are supported by the reported data. Figure 1, section I reports CFUs of *M. abscessus* at Day 1 and Day 3, while the paper states mice were sacrificed and organs harvested at Day 3 and Day 7. Is this an error in the figure, or was the Day 7 data not reported? Also, the paper states that 'livers' were harvested and homogenized, while the figure refers to 'lungs' - was this an error?*

Response: We greatly appreciate the comprehensive and insightful comments from you. We sincerely apologize for the significant errors in the original manuscript, particularly regarding the description of Figure 1G and the methods section.

The bacterial load data should have been correctly stated as being from Day 3 and Day 7, and the organs harvested were the lungs, not the liver (**Fig. RS9**). These errors were due to oversight during manuscript preparation. We deeply regret this mistake and have carefully revised both the figure and the corresponding text in the methods section to reflect the correct data.

Additionally, to improve clarity and provide additional context, we have updated Figure 1G (**now Figure 2A-B in the revised manuscript, Line 621-630**) and included a timeline graph to illustrate the experimental protocol and sampling schedule. We hope these corrections and additional details adequately address your concerns.

Thank you for pointing out this important issue, and we appreciate your patience and understanding as we improve the accuracy and quality of our manuscript.

Models.

2) Furthermore, there does not appear to be a statistically significant difference in CFUs between the Bedaquiline group and the combined Curcumin-Bedaquiline group in this figure.

Response: Thank you for this valuable comment. We understand your concern about the absence of statistical significance in CFU counts between the BDQ and BDQ-CUR groups as presented in the figure.

This outcome may be due to individual variations among mice, which can introduce variability in drug absorption after gavage. Such differences caused significant deviations and outliers in the lung bacterial load data. To ensure transparency, we included all raw data, which accounted for the absence of statistical significance in the analysis. However, upon closer examination, we observed a consistent trend within the BDQ-CUR group, where the bacterial load in the lungs was notably reduced compared to the untreated group. To address this issue, we have revised the description in the manuscript to provide a more precise interpretation of the data (**Line 127-130**):

Compared to untreated groups, BDQ-CUR (6/6), BDQ (3/6), and CUR (2/6) treatments reduced CFU counts by more than an order of magnitude (**Table 1**). HE staining demonstrated a reduction in inflammatory pathology in the lungs of BDQ-CUR-treated mice compared to untreated mice (Fig. 2C).

Additionally, we have included a new Table 1 in the revised manuscript to further illustrate the differences between the treatment and the control group. This table provides a clearer representation of the observed trends and offers additional context to support the conclusions. Below is a copy of the updated Table 1 for your reference.

Table 1. Summary of lung and spleen bacterial loads in immunocompetent and immunosuppressed mice after treatment (spleen samples collected only in immunosuppressed mice).

Mouse model	Organ	Treatment Groups	No. of improved infected mice (3 dpi) ^a	Mean difference from untreated group(log 10) ^b	Mean difference from initial infection(log 10)	No. of improved infected mice (7 dpi) ^a	Mean difference from untreated group(log 10) ^b	Mean difference from initial infection(log 10) ^b
Immunology normal	Lungs	CUR	0/6	0.74	/	2/6	0.67	/
		BDQ	1/6	0.54	/	3/6	1.23	/
		CUR-BDQ	3/6	1.25	/	6/6	1.77	/
Immunology suppressed	Lungs	CUR	0/3	-0.01	-1.06	0/4	0.68	-0.72
		BDQ	0/3	0.13	-0.91	0/4	1.36	-0.03
		CUR-BDQ	0/3	1.03	-0.02	4/4	3.22	1.83
	Spleen	CUR	0/3	0.03	-2.61	0/4	1.52	-1.31
		BDQ	0/3	0.27	-2.37	0/4	1.32	-1.51
		CUR-BDQ	0/3	1.46	-1.18	1/4	3.54	0.71

Note:

a: Immunology normal mice model: For mice treated with BDQ, CUR, or the CUR-BDQ combination, *No. of infection improved mice* refers to the number of mice whose individual bacterial load decreased by at least 1 log₁₀ unit compared to the average bacterial load of the **control group**, indicating an improvement in infection.

Immunology suppressed mice model: In this model, *No. of infection improved mice* refers to the number of mice whose individual bacterial load decreased by at least 1 log₁₀ unit compared to the average bacterial load of the **initial infection group**, indicating an improvement in infection.

b: Mean difference from untreated group or initial infection: Calculated as the average bacterial load of the control or initial infection group minus the average bacterial load of the treatment group. A positive value indicates a reduction in the treatment group's bacterial load compared to the control or initial infection group; a negative value indicates an increase. The CUR-BDQ combination group is highlighted in bold to emphasize its effectiveness.

We hope this revised description and additional data provide a more comprehensive response to your concerns. Thank you for your valuable feedback, which has significantly improved the clarity and accuracy of our manuscript.

3)If pathology cross sections of lungs are available, and can be analyzed quantitatively for the presence of infection/lesions, this could help further the claims of the combined treatment being more effective.

Response: Thank you for the reviewer's constructive and insightful comments. To strengthen our claims regarding the efficacy of the BDQ-CUR combination treatment, we have included pathological cross sections of lungs in the revised manuscript (**Fig. RS10**).

HE staining revealed a notable reduction in inflammatory pathology in the lungs of BDQ-CUR-treated mice compared to untreated mice. This observation provides additional evidence supporting the enhanced therapeutic effect of the combined treatment. The revised figure is now presented as **Fig. 2C** and **Line129-130** in the revised manuscript.

Fig. RS10. H&E staining of lung tissues from immunocompetent mice 7 days post-treatment.

We hope this additional data helps address your concern and provides stronger support for our conclusions. Thank you again for this valuable suggestion.

4)I would recommend that this in-vivo model be repeated in mice with a longer-term model of infection in C57Bl/6 mice or a novel immunocompromised mouse strain in which persistent or increasing infection can be demonstrated.

Response: Thank you for this constructive and insightful suggestion. Based on previous reports, we established an immunosuppressed mouse model using cyclophosphamide to replicate and further validate our findings in the mouse treatment study. In the revised manuscript, we have incorporated results from both the immunocompetent and immunosuppressed mouse infection models (**Fig. RS11**). Below is a summary of the updated findings:

Revised Description in the Manuscript (Line125-148 and Fig. 2)

Fig. 2

Fig. RS11. Curcumin Enhances Bedaquiline's Efficacy Against *M. abscessus* in Immunocompetent and Immunosuppressed Mouse Models.

As shown in Fig.2A timeline and in Fig. 2B results, BDQ-CUR could reduce the bacterial load in the lungs of normal C57BL/6 mice by day 7 of treatment. Compared to the untreated groups, BDQ-CUR (6/6), BDQ (3/6), and CUR (2/6) treatments

reduced CFU counts by more than an order of magnitude (Table 1). HE staining revealed a reduction in inflammatory pathology in the lungs of BDQ-CUR-treated mice compared to untreated mice (Fig. 2C).

Furthermore, similar to the intranasal infection model, immunodeficient mice develop a persistent infection that reaches the spleen (systemic)[31], [32]. We evaluated the effect of the combination using immunodeficient mice injected with cyclophosphamide (Fig. 2D)[33], [34]. The results showed that after 3 days of treatment, bacterial load in the BDQ-CUR groups was the same as the initial infection, while in other treatment groups (both the BDQ and CUR groups), the bacterial load significantly increased (Fig. 2E). The bacterial load in the spleen on day 3 in the combination group was also lower than that in the other groups, although it showed a slight increase relative to the initial infection (Fig. 2F). On day 7 of the combination treatment, compared with the BDQ group, both the lung and spleen bacterial loads significantly decreased by nearly 2 log₁₀ CFU (Fig. 2E and Fig. 2F). Similarly, compared to the initial infection, the combination treatment resulted in a significant reduction in bacterial loads by nearly 2 log₁₀ CFU in the lungs (4/4) and spleen (1/4) (Table 1). Pathological analysis on day 7 showed a reduction in the number of lymphocytes in these lesions, with moderate to minimal localized/focal histological features of inflammation observed in the BDQ-CUR groups. Severe and multifocal inflammation marked by the presence of lymphocytes and macrophages was observed in the lungs and spleen of the untreated group, in addition to extensive tissue damage (Fig. 2G). These findings support the reliability and effectiveness of the BDQ-CUR combination.

Conclusion

We appreciate your recommendation to evaluate the treatment in a longer-term infection model. The inclusion of data from the immunosuppressed mouse model has strengthened the evidence for the BDQ-CUR combination's efficacy. The updated results and analyses are now reflected in the revised manuscript (**Fig. 2 and Table 1**).

Reviewer 2:

To complete the peer review some vital aspects are missing in your methodology section. Please provide a detailed description of how you prepared the curcumin and bedaquiline samples for all experiments. Since both compounds are poor-water soluble the vehicles are extremely important for the results interpretation. The in vivo study is poorly described methodologically.

Response: Thank you for your constructive feedback. We sincerely apologize for the incomplete descriptions in the original manuscript and for any confusion caused by the highlighted areas. We have thoroughly revised the Materials and Methods section and made substantial improvements to the highlighted parts, including the Abstract and Importance sections, as detailed below:

1. Thorough Revision of Abstract

Line12-22:

In this study, we describe the combined effects of the important drug bedaquiline (BDQ) and the natural product curcumin (CUR) on *Mycobacterium abscessus*. In both in vitro and in vivo experiments, CUR enhanced BDQ's inhibitory effect. This combination reduced *M. abscessus* survival under nutrient-deprived, hypoxic, and acidic conditions, accelerated ATP depletion, mitigated BDQ-induced respiratory compensation, and effectively improved infection outcomes in both normal and immunosuppressed mice. Metabolomics analysis revealed that adding CUR to BDQ exacerbated BDQ-dependent downregulation of purine and pyrimidine metabolism and amino acid synthesis. Thus, BDQ-CUR combination therapy could potentially be applied to treat *M. abscessus* infections.

2. Substantial Updates to Importance Section

Line23-32:

Mycobacterium abscessus is an emerging pathogen that causes pulmonary infections, particularly in immunocompromised patients. It exhibits natural resistance to many anti-TB drugs, posing significant challenges for both patients and physicians, thereby raising the need for innovative drug discovery. Here, we describe the combined effects of the important drug bedaquiline (BDQ) and the natural product curcumin (CUR) on *M. abscessus*. In vitro and in vivo studies have shown that CUR enhances the inhibitory effect of BDQ. Additionally, we investigated the synergistic effects at the metabolic level. Thus, these findings highlight the potential of BDQ-CUR combination therapy against *M. abscessus* infections.

3. New Section: Bacterial Strains, Media, and Reagents

To address the methodology gaps highlighted by the reviewer, we added a new subsection titled “Bacterial Strains, Media, and Reagents” to the Materials and Methods section. This subsection provides essential details about the preparation of BDQ and CUR, including solvent use (DMSO) and sterilization methods.

Line271-273:

BDQ and CUR were dissolved in DMSO, while KM was dissolved in ultrapure water. All solutions were sterilized using a 0.22- μ m filter before use.

4. Expanded In Vivo Methodology

We thoroughly revised the in vivo study description to include comprehensive details for both the immunocompetent and immunosuppressed mouse models. These updates cover infection preparation, treatment regimens, sample collection, and data analysis, as summarized in the revised Materials and Methods section.

Line359-384:

M. abscessus infection was prepared by diluting the bacterial culture with PBS to a final volume of 20 μ L per mouse. A total of 54 immunocompetent C57BL/6 mice were intranasally inoculated with *M. abscessus* ($\sim 1 \times 10^9$ CFU/mouse). After 1 day, the infected mice were divided randomly into four

treatment groups as follows: Group 1 received DMSO only as a control. Group 2 was treated with BDQ at a dose of 30 mg/kg. Group 3 received CUR at a dose of 200 mg/kg. Group 4 was treated with a combination of BDQ (30 mg/kg) and CUR (200 mg/kg). All treatments were prepared in DMSO and administered by daily oral gavage for 1 week.

On day 1 post-infection, six mice were sacrificed to determine the initial bacterial load in the lungs. On days 3 or 7 post-treatment, six mice of each group were sacrificed. The upper lobes of the right lungs were dissected, fixed in 4% paraformaldehyde for 24 hours, and sectioned for histological analysis using H&E staining. The remaining lung tissues were homogenized, serially diluted 10-fold, and plated on 7H10 agar. Bacterial load in the lungs was determined by counting colonies after 5 days of incubation at 37°C.

For the immunosuppression mouse infection, studies were conducted to mimic respiratory infection as described previously[67]. Briefly, 32 six-week-old female C57BL/6 mice were rendered neutropenic by intraperitoneal injection of cyclophosphamide at 150 mg/kg, administered 4 days and 1 day prior to infection. The mice were then infected intranasally with $\sim 1 \times 10^7$ CFU of *M. abscessus*. On day 3 post-infection, four mice were sacrificed to determine the initial bacterial load in the lungs and spleens. The remaining 28 infected mice were divided randomly into the same four treatment groups as the immunocompetent mice. Three or four mice from each group were sacrificed on days 3 or 7 post-treatment. Lungs and spleens were collected for CFU count and histological analysis using H&E staining.

We hope these revisions adequately address your concerns regarding the highlighted areas and methodology. In addition to the specific sections mentioned, we have also carefully reviewed and refined other parts of the Materials and Methods section to ensure consistency, clarity, and accuracy throughout the manuscript. Thank you for your valuable suggestions, which have greatly enhanced the clarity and rigor of our work.

References

- [1] E. Marini *et al.*, “Curcumin, an antibiotic resistance breaker against a multiresistant clinical isolate of *Mycobacterium abscessus*,” *Phytother. Res. PTR*, vol. 32, no. 3, pp. 488–495, Mar. 2018, doi: 10.1002/ptr.5994.
- [2] N. Ahmad *et al.*, “Treatment correlates of successful outcomes in pulmonary multidrug-resistant tuberculosis: an individual patient data meta-analysis,” *The Lancet*, vol. 392, no. 10150, pp. 821–834, Sep. 2018, doi: 10.1016/S0140-6736(18)31644-1.
- [3] M. Gao *et al.*, “Early outcome and safety of bedaquiline-containing regimens for treatment of MDR- and XDR-TB in China: a multicentre study,” *Clin. Microbiol. Infect.*, vol. 27, no. 4, pp. 597–602, Apr. 2021, doi: 10.1016/j.cmi.2020.06.004.
- [4] A. S. Pym *et al.*, “Bedaquiline in the treatment of multidrug- and extensively drug-resistant tuberculosis,” *Eur. Respir. J.*, vol. 47, no. 2, pp. 564–574, Jan. 2016, doi: 10.1183/13993003.00724-2015.
- [5] H. Patel, R. Pawara, K. Pawara, F. Ahmed, A. Shirkhedkar, and S. Surana, “A structural insight of bedaquiline for the cardiotoxicity and hepatotoxicity,” *Tuberculosis*, vol. 117, pp. 79–84, Jul. 2019, doi: 10.1016/j.tube.2019.06.005.
- [6] U. M. Hanumegowda, G. Wenke, A. Regueiro-Ren, R. Yordanova, J. P. Corradi, and S. P. Adams, “Phospholipidosis as a Function of Basicity, Lipophilicity, and Volume of Distribution of Compounds,” *Chem. Res. Toxicol.*, vol. 23, no. 4, pp. 749–755, Apr. 2010, doi: 10.1021/tx9003825.
- [7] A. L. Mulyukin *et al.*, “Distinct Effects of Moxifloxacin and Bedaquiline on Growing and ‘Non-Culturable’ *Mycobacterium abscessus*,” *Microorganisms*, vol. 11, no. 11, p. 2690, Nov. 2023, doi: 10.3390/microorganisms11112690.

- [8] O. Martins, J. Lee, A. Kaushik, N. C. Ammerman, K. E. Dooley, and E. L. Nuermberger, "In Vitro Activity of Bedaquiline and Imipenem against Actively Growing, Nutrient-Starved, and Intracellular *Mycobacterium abscessus*," *Antimicrob. Agents Chemother.*, vol. 65, no. 12, p. e0154521, Nov. 2021, doi: 10.1128/AAC.01545-21.
- [9] J. Lee, N. Ammerman, A. Agarwal, M. Naji, S.-Y. Li, and E. Nuermberger, "Differential In Vitro Activities of Individual Drugs and Bedaquiline-Rifabutin Combinations against Actively Multiplying and Nutrient-Starved *Mycobacterium abscessus*," *Antimicrob. Agents Chemother.*, vol. 65, no. 2, pp. e02179-20, Jan. 2021, doi: 10.1128/AAC.02179-20.
- [10] A. Lanni *et al.*, "Activity of Drug Combinations against *Mycobacterium abscessus* Grown in Aerobic and Hypoxic Conditions," *Microorganisms*, vol. 10, no. 7, p. 1421, Jul. 2022, doi: 10.3390/microorganisms10071421.
- [11] N. Li *et al.*, "Novel dissymmetric 3,5-bis(arylidene)-4-piperidones as potential antitumor agents with biological evaluation *in vitro* and *in vivo*," *Eur. J. Med. Chem.*, vol. 147, pp. 21–33, Mar. 2018, doi: 10.1016/j.ejmech.2018.01.088.
- [12] C.-W. Yang, C.-L. Chang, H.-C. Lee, C.-W. Chi, J.-P. Pan, and W.-C. Yang, "Curcumin induces the apoptosis of human monocytic leukemia THP-1 cells via the activation of JNK/ERK pathways," *BMC Complement. Altern. Med.*, vol. 12, p. 22, Mar. 2012, doi: 10.1186/1472-6882-12-22.
- [13] S. M. M. Faudzi *et al.*, "Synthesis, biological evaluation and QSAR studies of diarylpentanoid analogues as potential nitric oxideinhibitors," *MedChemComm*, vol. 6, no. 6, pp. 1069–1080, Jun. 2015, doi: 10.1039/C4MD00541D.
- [14] N. Barua and A. K. Buragohain, "Therapeutic Potential of Curcumin as an Antimycobacterial Agent," *Biomolecules*, vol. 11, no. 9, p. 1278, Aug. 2021, doi: 10.3390/biom11091278.
- [15] J. S. Mackenzie *et al.*, "Bedaquiline reprograms central metabolism to reveal glycolytic vulnerability in *Mycobacterium tuberculosis*," *Nat. Commun.*, vol. 11, no. 1, p. 6092, Nov. 2020, doi: 10.1038/s41467-020-19959-4.

Re: Spectrum02295-24R1 (**Curcumin Enhances Bedaquiline's Efficacy Against *Mycobacterium abscessus*: In Vitro and In Vivo Evidence**)

Dear Prof. Zhe Wang:

Thank you for the privilege of reviewing your work. Below you will find my comments, instructions from the Spectrum editorial office, and the reviewer comments.

Please address the Reviewer's final comment, including the DMSO concentration in the oral preparations. Also please add all figures in your Response to Reviewers' that were not added to the manuscript itself as supplemental material. Finally, if the difference in CFU is not statistically significant you must explicitly state that in the manuscript (ie you can use the word 'trend' etc).

Revision Guidelines

Sincerely,
Kayvan Zainabadi
Editor
Microbiology Spectrum

Reviewer #2 (Comments for the Author):

Thank you for all the clarifications; they have substantially improved the manuscript.

As a final suggestion, could you please include the final DMSO concentration in all sections where DMSO was used as a solvent? This will help ensure clarity and consistency throughout the manuscript.

Thank you once again for your hard work on this revision.

Reviewer 2:

To complete the peer review some vital aspects are missing in your methodology section. Please provide a detailed description of how you prepared the curcumin and bedaquiline samples for all experiments. Since both compounds are poor-water soluble the vehicles are extremely important for the results interpretation. The in vivo study is poorly described methodologically.

Response: Thank you for your constructive feedback. We sincerely apologize for the incomplete descriptions in the original manuscript and for any confusion caused by the highlighted areas. We have thoroughly revised the Materials and Methods section and made substantial improvements to the highlighted parts, including the Abstract and Importance sections, as detailed below:

1. Thorough Revision of Abstract

Line12-22:

In this study, we describe the combined effects of ~~the important drug~~ bedaquiline (BDQ) and the natural product curcumin (CUR) on *Mycobacterium abscessus*. In both in vitro and in vivo experiments, CUR enhanced BDQ's inhibitory effect. This combination reduced *M. abscessus* survival under nutrient-deprived, hypoxic, and acidic conditions, accelerated ATP depletion, mitigated BDQ-induced respiratory compensation, and effectively improved infection outcomes in both normal and immunosuppressed mice. Metabolomics analysis revealed that adding CUR to BDQ exacerbated BDQ-dependent downregulation of purine and pyrimidine metabolism and amino acid synthesis. Thus, BDQ-CUR combination therapy could potentially be applied to treat *M. abscessus* infections.

2. Substantial Updates to Importance Section

Line23-32:

Mycobacterium abscessus is an emerging pathogen that causes pulmonary infections, particularly in immunocompromised patients. It exhibits natural

resistance to many anti-TB drugs, posing significant challenges for both patients and physicians, thereby raising the need for innovative drug discovery. Here, we describe the combined effects of ~~the important drug bedaquiline (BDQ) and the natural product curcumin (CUR)~~ on *M. abscessus*. In vitro and in vivo studies have shown that CUR enhances the inhibitory effect of BDQ. Additionally, we investigated the synergistic effects at the metabolic level. Thus, these findings highlight the potential of BDQ-CUR combination therapy against *M. abscessus* infections.

3. New Section: Bacterial Strains, Media, and Reagents

To address the methodology gaps highlighted by the reviewer, we added a new subsection titled “Bacterial Strains, Media, and Reagents” to the Materials and Methods section. This subsection provides essential details about the preparation of BDQ and CUR, including solvent use (DMSO) and sterilization methods.

Line271-273:

BDQ and CUR were dissolved in DMSO **Please include the final concentration of DMSO (%) in all preparations**, while KM was dissolved in ultrapure water. All solutions were sterilized using a 0.22- μ m filter before use.

4. Expanded In Vivo Methodology

We thoroughly revised the in vivo study description to include comprehensive details for both the immunocompetent and immunosuppressed mouse models. These updates cover infection preparation, treatment regimens, sample collection, and data analysis, as summarized in the revised Materials and Methods section.

Line359-384:

M. abscessus infection was prepared by diluting the bacterial culture with PBS to a final volume of 20 μ L per mouse. A total of 54 immunocompetent C57BL/6 mice were intranasally inoculated with *M. abscessus* ($\sim 1 \times 10^9$ CFU/mouse). After 1 day, the infected mice were divided randomly into four treatment groups as follows: Group 1 received DMSO only as a control **add the**

final volume here. Group 2 was treated with BDQ at a dose of 30 mg/kg. Group 3 received CUR at a dose of 200 mg/kg. Group 4 was treated with a combination of BDQ (30 mg/kg) and CUR (200 mg/kg). All treatments were prepared in DMSO add here the final amount in % of each treatment and administered by daily oral gavage for 1 week.

On day 1 post-infection, six mice were sacrificed to determine the initial bacterial load in the lungs. On days 3 or 7 post-treatment, six mice of each group were sacrificed. The upper lobes of the right lungs were dissected, fixed in 4% paraformaldehyde for 24 hours, and sectioned for histological analysis using H&E staining. The remaining lung tissues were homogenized, serially diluted 10-fold, and plated on 7H10 agar. Bacterial load in the lungs was determined by counting colonies after 5 days of incubation at 37°C.

For the immunosuppression mouse infection, studies were conducted to mimic respiratory infection as described previously[67]. Briefly, 32 six-week-old female C57BL/6 mice were rendered neutropenic by intraperitoneal injection of cyclophosphamide at 150 mg/kg, administered 4 days and 1 day prior to infection. The mice were then infected intranasally with $\sim 1 \times 10^7$ CFU of *M. abscessus*. On day 3 post-infection, four mice were sacrificed to determine the initial bacterial load in the lungs and spleens. The remaining 28 infected mice were divided randomly into the same four treatment groups as the immunocompetent mice. Three or four mice from each group were sacrificed on days 3 or 7 post-treatment. Lungs and spleens were collected for CFU count and histological analysis using H&E staining.

We hope these revisions adequately address your concerns regarding the highlighted areas and methodology. In addition to the specific sections mentioned, we have also carefully reviewed and refined other parts of the Materials and Methods section to ensure consistency, clarity, and accuracy throughout the manuscript. Thank you for your valuable suggestions, which have greatly enhanced the clarity and rigor of our work.

References

- [1] E. Marini *et al.*, “Curcumin, an antibiotic resistance breaker against a multiresistant clinical isolate of *Mycobacterium abscessus*,” *Phytother. Res. PTR*, vol. 32, no. 3, pp. 488–495, Mar. 2018, doi: 10.1002/ptr.5994.
- [2] N. Ahmad *et al.*, “Treatment correlates of successful outcomes in pulmonary multidrug-resistant tuberculosis: an individual patient data meta-analysis,” *The Lancet*, vol. 392, no. 10150, pp. 821–834, Sep. 2018, doi: 10.1016/S0140-6736(18)31644-1.
- [3] M. Gao *et al.*, “Early outcome and safety of bedaquiline-containing regimens for treatment of MDR- and XDR-TB in China: a multicentre study,” *Clin. Microbiol. Infect.*, vol. 27, no. 4, pp. 597–602, Apr. 2021, doi: 10.1016/j.cmi.2020.06.004.
- [4] A. S. Pym *et al.*, “Bedaquiline in the treatment of multidrug- and extensively drug-resistant tuberculosis,” *Eur. Respir. J.*, vol. 47, no. 2, pp. 564–574, Jan. 2016, doi: 10.1183/13993003.00724-2015.
- [5] H. Patel, R. Pawara, K. Pawara, F. Ahmed, A. Shirkhedkar, and S. Surana, “A structural insight of bedaquiline for the cardiotoxicity and hepatotoxicity,” *Tuberculosis*, vol. 117, pp. 79–84, Jul. 2019, doi: 10.1016/j.tube.2019.06.005.
- [6] U. M. Hanumegowda, G. Wenke, A. Regueiro-Ren, R. Yordanova, J. P. Corradi, and S. P. Adams, “Phospholipidosis as a Function of Basicity, Lipophilicity, and Volume of Distribution of Compounds,” *Chem. Res. Toxicol.*, vol. 23, no. 4, pp. 749–755, Apr. 2010, doi: 10.1021/tx9003825.
- [7] A. L. Mulyukin *et al.*, “Distinct Effects of Moxifloxacin and Bedaquiline on Growing and ‘Non-Culturable’ *Mycobacterium abscessus*,” *Microorganisms*, vol. 11, no. 11, p. 2690, Nov. 2023, doi: 10.3390/microorganisms11112690.

- [8] O. Martins, J. Lee, A. Kaushik, N. C. Ammerman, K. E. Dooley, and E. L. Nuermberger, "In Vitro Activity of Bedaquiline and Imipenem against Actively Growing, Nutrient-Starved, and Intracellular *Mycobacterium abscessus*," *Antimicrob. Agents Chemother.*, vol. 65, no. 12, p. e0154521, Nov. 2021, doi: 10.1128/AAC.01545-21.
- [9] J. Lee, N. Ammerman, A. Agarwal, M. Naji, S.-Y. Li, and E. Nuermberger, "Differential In Vitro Activities of Individual Drugs and Bedaquiline-Rifabutin Combinations against Actively Multiplying and Nutrient-Starved *Mycobacterium abscessus*," *Antimicrob. Agents Chemother.*, vol. 65, no. 2, pp. e02179-20, Jan. 2021, doi: 10.1128/AAC.02179-20.
- [10] A. Lanni *et al.*, "Activity of Drug Combinations against *Mycobacterium abscessus* Grown in Aerobic and Hypoxic Conditions," *Microorganisms*, vol. 10, no. 7, p. 1421, Jul. 2022, doi: 10.3390/microorganisms10071421.
- [11] N. Li *et al.*, "Novel dissymmetric 3,5-bis(arylidene)-4-piperidones as potential antitumor agents with biological evaluation *in vitro* and *in vivo*," *Eur. J. Med. Chem.*, vol. 147, pp. 21–33, Mar. 2018, doi: 10.1016/j.ejmech.2018.01.088.
- [12] C.-W. Yang, C.-L. Chang, H.-C. Lee, C.-W. Chi, J.-P. Pan, and W.-C. Yang, "Curcumin induces the apoptosis of human monocytic leukemia THP-1 cells via the activation of JNK/ERK pathways," *BMC Complement. Altern. Med.*, vol. 12, p. 22, Mar. 2012, doi: 10.1186/1472-6882-12-22.
- [13] S. M. M. Faudzi *et al.*, "Synthesis, biological evaluation and QSAR studies of diarylpentanoid analogues as potential nitric oxideinhibitors," *MedChemComm*, vol. 6, no. 6, pp. 1069–1080, Jun. 2015, doi: 10.1039/C4MD00541D.
- [14] N. Barua and A. K. Buragohain, "Therapeutic Potential of Curcumin as an Antimycobacterial Agent," *Biomolecules*, vol. 11, no. 9, p. 1278, Aug. 2021, doi: 10.3390/biom11091278.
- [15] J. S. Mackenzie *et al.*, "Bedaquiline reprograms central metabolism to reveal glycolytic vulnerability in *Mycobacterium tuberculosis*," *Nat. Commun.*, vol. 11, no. 1, p. 6092, Nov. 2020, doi: 10.1038/s41467-020-19959-4.

Dear Editors and Reviewers:

We sincerely thank the reviewers for your thoughtful and constructive comments, which have greatly improved the quality of our manuscript (**Spectrum 02295-24**). Below, we provide point-by-point responses to address each comment, accompanied by the corresponding revisions in the manuscript.

Point-by-point Responses

Reviewer 2:

As a final suggestion, could you please include the final DMSO concentration in all sections where DMSO was used as a solvent? This will help ensure clarity and consistency throughout the manuscript.

Response:

Thank you for your constructive suggestion. We have thoroughly reviewed the manuscript and ensured that the final DMSO concentration is explicitly stated in all relevant sections where DMSO was used as a solvent. Specifically, the DMSO concentration used in all experiments was 20% (v/v). For example, the updated text in the Materials and Methods section now reads:

BDQ and CUR were dissolved in a solution containing 20% (v/v) DMSO in water.

This revision ensures clarity and consistency throughout the manuscript. All relevant updates have been highlighted in the revised manuscript for your convenience.

Additional Notes:

Please address the Reviewer's final comment, including the DMSO concentration in the oral preparations.

Response:

We have added detailed descriptions of the DMSO concentration throughout the manuscript as requested.

Also please add all figures in your Response to Reviewers' that were not added to the manuscript itself as supplemental material.

Response:

Thank you for your comment. We have incorporated all figures from our previous response, including the results of the in vitro phenotypic assessment and cytotoxicity determination, as supplemental material. These figures, along with their corresponding legends, have been compiled into a single file titled **Supplemental Data.docx** for ease of review and access.

Finally, if the difference in CFU is not statistically significant you must explicitly state that in the manuscript (ie you can use the word 'trend' etc).

Response:

Thank you for your comment. We have explicitly revised the manuscript to state that the differences in CFU observed in the normal immune mice were not statistically significant. Terms such as "trend" have been used appropriately to describe the results. The revised text in the Results section (**lines 130-133**) reads as follows:

Although compared to BDQ alone, the difference in lung bacterial load in the combined treatment group was not statistically significant, the lung bacterial load showed a trend of 0.5 log₁₀ unit reduction, and pathological sections showed significant improvement (Fig. 2B and Fig. 2C).

Re: Spectrum02295-24R2 (**Curcumin Enhances Bedaquiline's Efficacy Against *Mycobacterium abscessus*: In Vitro and In Vivo Evidence**)

Dear Prof. Zhe Wang:

Your manuscript has been accepted, and I am forwarding it to the ASM production staff for publication. Your paper will first be checked to make sure all elements meet the technical requirements. ASM staff will contact you if anything needs to be revised before copyediting and production can begin. Otherwise, you will be notified when your proofs are ready to be viewed.

Sincerely,
Kayvan Zainabadi
Editor
Microbiology Spectrum